# Learnable Burst-Encodable Time-of-Flight Imaging for High-Fidelity Long-Distance Depth Sensing

**Manchao Bao[1]**
manchaobao@smail.nju.edu.cn

**Shengjiang Fang[1,2]**
shengjiangfang@smail.nju.edu.cn

**Tao Yue[1]**
yuetao@nju.edu.cn

**Xuemei Hu[1]**
xuemeihu@nju.edu.cn

[1]Nanjing University

[2]Nanjing Electronic Devices Institute

## Abstract

Long-distance depth imaging holds great promise for applications such as autonomous driving and robotics. Direct time-of-flight (dToF) imaging offers high-precision, long-distance depth sensing, yet demands ultra-short pulse light sources and high-resolution time-to-digital converters. In contrast, indirect time-of-flight (iToF) imaging often suffers from phase wrapping and low signal-to-noise ratio (SNR) as the sensing distance increases. In this paper, we introduce a novel ToF imaging paradigm, termed Burst-Encodable Time-of-Flight (BE-ToF), which facilitates high-fidelity, long-distance depth imaging. Specifically, the BE-ToF system emits light pulses in burst mode and estimates the phase delay of the reflected signal over the entire burst period, thereby effectively avoiding the phase wrapping inherent to conventional iToF systems. Moreover, to address the low SNR caused by light attenuation over increasing distances, we propose an end-to-end learnable framework that jointly optimizes the coding functions and the depth reconstruction network. A specialized double well function and first-order difference term are incorporated into the framework to ensure the hardware implementability of the coding functions. The proposed approach is rigorously validated through comprehensive simulations and real-world prototype experiments, demonstrating its effectiveness and practical applicability. The code is available at: https://github.com/ComputationalPerceptionLab/BE-ToF.

## 1 Introduction

Achieving high-precision depth imaging over long distances has remained a fundamental objective in fields such as computer vision, robotics, and autonomous systems. Time-of-flight (ToF) imaging [1, 2, 3], as a key approach to depth imaging, can be further categorized into direct ToF (dToF) and indirect ToF (iToF) based on differences in working principles. Direct ToF imaging [4] estimates depth by directly measuring the round-trip time of light, enabling high-precision and long-range sensing. Despite its advantages, this approach requires ultra-short pulsed light sources and high-resolution time-to-digital converters (TDCs), imposing stringent hardware demands that increase system complexity and cost, thereby limiting its practicality for widespread deployment. Indirect ToF systems [5, 6, 7, 8, 9], in contrast, emit amplitude-modulated continuous wave (AMCW) signals and infer depth by analyzing the phase shift between the transmitted and received signals. Due to their relatively lower hardware complexity and cost, iToF systems offer a more practical and hardware-friendly solution. Nevertheless, existing iToF technologies face significant challenges in long-range imaging, primarily due to phase wrapping [10] and low signal-to-noise ratio (SNR) resulting from optical attenuation [11]. To address the phase wrapping, dual-frequency modulation

techniques [12, 13] have been proposed, albeit at the cost of increased computational complexity and stricter hardware synchronization requirements. Alternative approaches have sought to mitigate phase wrapping under single-frequency modulation by incorporating scene priors [5, 14], however, these methods do not fundamentally resolve the intrinsic ambiguity introduced by periodic modulation.

In this paper, we propose a novel ToF imaging paradigm termed Burst-Encodable Time-of-Flight (BE-ToF). Our BE-ToF system operates in a low-frequency burst mode for light pulse modulation and demodulation, such that the phase of the reflected signal sweeps the entire range $[0, 2\pi]$ within a single, long burst period. This facilitates high-fidelity, long-distance depth imaging using only single frequency modulation. Moreover, considering the significant variation in SNRs caused by the light-falloff, we propose an end-to-end learnable framework that jointly optimizes the coding functions and the depth reconstruction network, thereby ensuring high-precision depth estimation. In particular, we incorporate constraints based on double well function and first-order difference to ensure the hardware implementability of the learned coding functions. We evaluate our method on a synthetic dataset and compare it with conventional iToF approaches, including single-frequency and multi-frequency modulation techniques. Finally, we built a prototype system to prove the effectiveness of our method in real-world experiments.

In general, we make the following contributions:

- We present a novel Burst-Encodable Time-of-Flight imaging system that enables high-fidelity long-distance depth sensing using only a single modulation frequency, thereby fundamentally mitigating the issue of phase wrapping inherent in traditional iToF systems.

- We propose an end-to-end learnable framework that jointly optimizes the coding functions and the depth reconstruction network to ensure high-precision depth estimation across varying distances.

- We uniquely incorporate double well function and first-order difference as loss function to ensure the hardware implementability of the learned coding functions.

- We develop a prototype of our BE-ToF system and demonstrate its superior performance on both synthetic datasets and real-world scenarios.

## 2   Related Work

**ToF imaging.**   Time-of-flight (ToF) imaging has become a widely adopted and effective modality for depth acquisition. Direct ToF (dToF) enables long-range depth estimation by measuring the round-trip time of short optical pulses [15]. However, attaining high precision with dToF places stringent demands on the illumination and timing hardware, typically requiring nanosecond- or even picosecond-scale pulse widths [4, 16, 17] as well as TDCs with tens-of-picoseconds resolution and low timing jitter [18, 19]. These requirements substantially hinder practical implementation and large-scale deployment. In contrast, indirect ToF (iToF) imaging leverages cost-effective CMOS sensors to achieve high-resolution depth estimation, yet it is inherently susceptible to phase ambiguity due to phase wrapping in long-range scenarios. A common strategy to alleviate this issue is multi-frequency modulation [12, 20, 21], where low modulation frequencies extend the maximum unambiguous range and high frequencies preserve depth precision. For example, Hanto *et al.* [22] developed a ToF LiDAR range finder based on dual-modulation frequency switching to extend the imaging range, and Su *et al.* [23] proposed an end-to-end ToF framework for high-quality depth reconstruction under multi-frequency modulation. Nevertheless, multi-frequency operation often increases hardware complexity and computational cost. Alternatively, single-frequency phase unwrapping has been explored using amplitude correction [5], surface normal constraints [14], and RGB fusion [24], but such approaches typically rely heavily on scene priors and may degrade under challenging conditions. In this paper, we propose Burst-Encodable Time-of-Flight Imaging (BE-ToF) to fundamentally address phase wrapping in iToF, enabling high-fidelity, long-distance depth estimation.

**End-to-end learning.**   End-to-end learning is a method aimed at jointly optimizing optical systems and reconstruction algorithms. Metzler *et al.* [25, 26] obtained high dynamic range (HDR) images from a single-shot by jointly optimizing the optical encoder and the electronic decoder. Nie *et al.* [27] leveraged an end-to-end network for hyperspectral reconstruction, enabling simultaneous

learning of optimized camera spectral response functions and a mapping for spectral reconstruction. For dense 3D localization microscopy, Nehme *et al.* [28] proposed a deep STORM-based method to achieve end-to-end optimization of point spread function engineering and accurate 3D localization. To achieve extended depth of field (EDOF), Sitzmann *et al.* [29] proposed to jointly optimize the optical system and the reconstruction algorithm's parameters to achieve achromatic EDOF imaging. Guo *et al.* [30] put forward an end-to-end framework capable of jointly optimizing the coding functions and the exposure time to improve the accuracy of fluorescence lifetime imaging. Moreover, in iToF imaging, Chugunov *et al.* [31] proposed to jointly learn a microlens amplitude mask and an encoder-decoder network to reduce flying pixels in depth captures. Li *et al.* [11] put forward a Fisher-information-guided framework for the joint optimization of the coding functions and the reconstruction network. Given the remarkable potential of end-to-end learning in elevating imaging performance, we propose an end-to-end learnable framework that jointly optimizes the coding functions and the depth reconstruction network of our BE-ToF, ensuring high-quality depth performance across varying distances.

## 3 Learnable Burst-Encodable Time-of-Flight Imaging

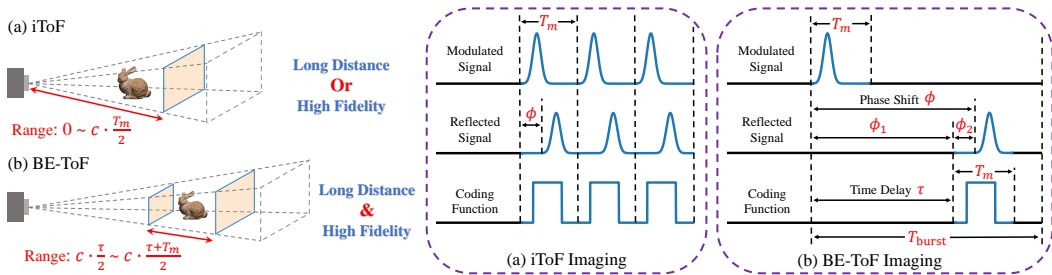

Figure 1: Comparison between iToF and BE-ToF. (a) Principle of iToF imaging, which suffers from a trade-off between sensing distance and precision; (b) Principle of BE-ToF imaging, enabling long-distance and high-fidelity depth sensing through modulation and demodulation in burst mode.

In this section, we first introduce the working principle of our BE-ToF. As shown in Fig. 1(a), conventional iToF is fundamentally constrained by a trade-off between maximum unambiguous range and depth precision, governed by the modulation period $T_m$. To handle this, our BE-ToF performs short-period light pulse modulation/demodulation in a low-frequency burst mode. As illustrated in Fig. 1(b), within each long burst period $T_{burst}$, a single modulated signal is emitted. When the reflected signal returns with a phase shift $\phi$, it can be demodulated by coding functions with controllable time delay $\tau$. Specifically, the total phase shift $\phi$ can be decomposed into two components: $\phi_1$, which is primarily determined by the controllable time delay $\tau$, and $\phi_2$, which can be recovered using demodulation techniques like 4-step phase shift [5] or deep learning [23, 11]. In summary, the depth $d$ can be defined as Eq. 1

$$d = \frac{c\,(\phi_1 + \phi_2)\,T_{\text{burst}}}{4\pi} = \frac{c\tau}{2} + \mathcal{D}(\phi_2)\,, \tag{1}$$

where $c$ is the light speed and $\mathcal{D}(\phi_2)$ represents the demodulation process of $\phi_2$.

Thus, in our BE-ToF system, the maximum unambiguous range $d_{mur}$ is primarily determined by burst period $T_{burst}$, as defined in Eq. 2

$$d_{mur} = \frac{c}{2f_{burst}} = \frac{c \cdot T_{burst}}{2}\,. \tag{2}$$

Regarding depth precision, since we divide the phase delay $\phi$ into two components $\phi_1$ and $\phi_2$, where $\phi_1$ is entirely determined by the time delay $\tau$. Consequently, the depth error in our BE-ToF system mainly arises during the demodulation of $\phi_2$. Thus, the depth error $\epsilon_d$ of our BE-ToF system can be represented as Eq. 3

$$\epsilon_d = \frac{c \cdot \epsilon_{\phi_2}}{4\pi f_m} = \frac{c \cdot \epsilon_{\phi_2} \cdot T_m}{4\pi}\,, \tag{3}$$

where $\epsilon_{\phi_2}$ is the phase error due to several factors like photon noise, readout noise, and multi-path interference. For fixed phase error, the depth error is chiefly governed by the modulation/demodulation period $T_m$.

Based on the above analysis, BE-ToF substantially extends the maximum unambiguous range while maintaining the same depth precision as conventional iToF, thereby enabling long distance and high-fidelity depth imaging. Moreover, the depth sensing range of our BE-ToF spans from $\frac{c \cdot \tau}{2}$ to $\frac{c \cdot (\tau + T_m)}{2}$, which can be flexibly adjusted by tuning the time delay $\tau$.

## 3.1 Differential BE-ToF Imaging Model

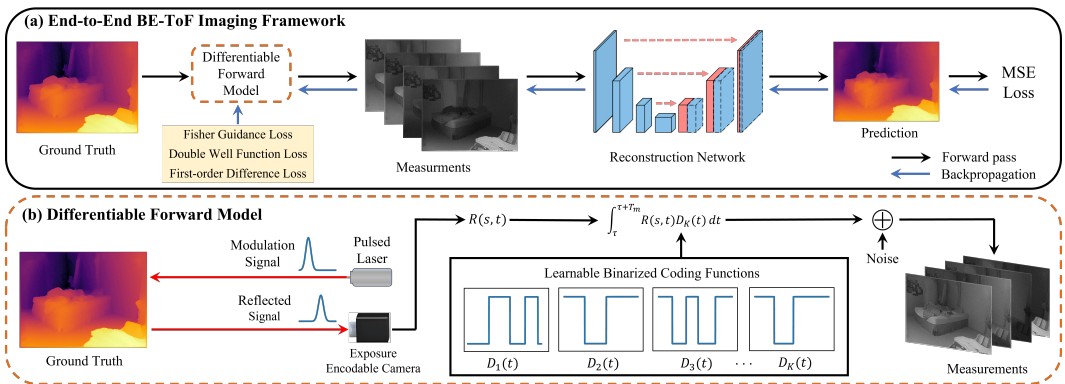

Figure 2: (a) End-to-end BE-ToF imaging framework for jointly optimizing the coding functions and reconstruction network, (b) The differentiable physical model of the BE-ToF system.

Based on the operating principle of BE-ToF, which can be realized with a pulsed laser and an exposure-encodable camera [32], we propose an end-to-end imaging framework to ensure high-quality reconstruction across varying distances and SNRs. As shown in Fig. 2(a), the framework comprises two components: a differentiable forward model that synthesizes BE-ToF measurements in our simulation pipeline, and a reconstruction network that estimates depth from multiple measurements. By jointly optimizing the coding functions and the reconstruction network, the system delivers accurate depth reconstructions under challenging conditions.

In this section, we first establish the differentiable forward model of our BE-ToF for end-to-end optimization, as illustrated in Fig. 2(b). Assuming $M(t)$ is the modulated signal emitted by pulse laser, the reflected signal of scene point $s \in \mathbb{R}^3$ can be defined as Eq. 4

$$R(s,t) = \rho_s M(t - 2\frac{d(s)}{c}) + I_{amb}, \tag{4}$$

where $\rho_s$ is the inherent reflectance of the scene point $s$, $I_{amb}$ is the ambient light, $d(s)$ denotes the depth value of point $s$. Furthermore, considering the attenuation of light intensity with distance during propagation, we incorporate the attenuation function into our model as Eq. 5

$$R(s,t) = \mathcal{F}_{d(s)}\rho_s M(t - 2\frac{d(s)}{c}) + I_{amb}, \tag{5}$$

where $\mathcal{F}_{d(s)}$ is the attenuation coefficient of the emitted light $M(t)$ at depth $d(s)$, which is typically inversely proportional to the square of the distance [33]. Finally, the whole BE-ToF imaging process can be formulated as Eq. 6

$$I_i(s) = \int_{\tau}^{\tau + T_m} R(s,t)D_i(t)\,dt, \quad i \in 1,...,K, \tag{6}$$

where $I_i(s)$ is the measurement value of the camera, $D_i(t)$ denotes the coding functions and $K$ denotes the number of measurements. Taking into account the inherent noise of the sensor, the final measurement can be expressed as Eq. 7

$$X_i(s) = I_i(s) + n_d + n_r, \quad n_d \sim \mathcal{P}(\mathbb{E}(n_d)), \ n_r \sim \mathcal{N}(0, \sigma_r^2), \tag{7}$$

where $n_d$ is the dark noise following the Poisson distribution with expectation $\mathbb{E}(n_d)$ and $n_r$ is the readout noise following Gaussian distribution with standard deviation $\sigma_r$.

Considering that $X_i(s)$ contains three unknowns: $\rho_s$, $I_{amb}$, $d(s)$. Therefore, at least $K \geq 3$ measurements are required to solve for the depth $d(s)$.

## 3.2   Reconstruction Network

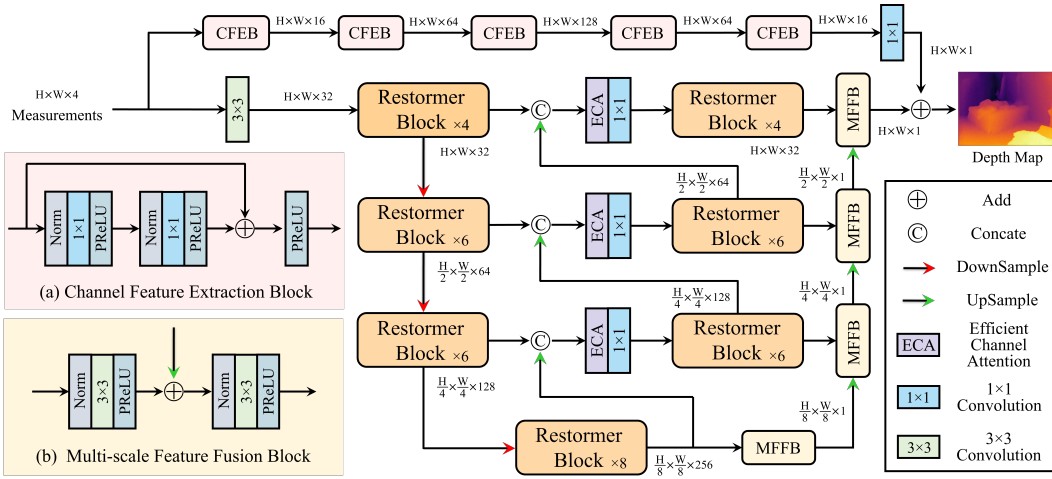

Figure 3: Architecture of the Restormer-based Spatial-Channel Fusion Network(RSCF-Net), (a) Channel Feature Extraction Block(CFEB), (b) Multi-scale Feature Fusion Block(MFFB).

With the proposed differentiable forward model, we can simulate the $K$ measurements of the BE-ToF imaging process. To recover high-fidelity depth map from this set of measurements, we propose a Restormer-based Spatial-Channel Fusion Network(RSCF-Net). As shown in Fig. 3, our network adopts Restormer [34] as the backbone, featuring a four-level encoder-decoder structure in which each level comprises multiple Restormer blocks. In contrast to the conventional skip connections used in the original Restormer, we integrate an Efficient Channel Attention (ECA) module [35] to enhance the fusion of features between encoder and decoder branches. Furthermore, recognizing the inherent differences between depth reconstruction and the image restoration tasks for which Restormer was originally designed, we augment our network with two additional components: the Channel Feature Extraction Block (CFEB) and the Multi-scale Feature Fusion Block (MFFB). The CFEB is composed of multiple residual-connected 1×1 convolutional layers, designed to extract inter-channel relationships across multiple per-pixel measurements. On the other hand, the MFFB emphasizes spatial structure by performing preliminary depth estimation at each decoder level and progressively integrating features from multiple scales in a coarse-to-fine manner. The outputs of CFEB and MFFB are subsequently fused to produce the final high-fidelity depth map.

## 3.3   Loss Function

During the training process, we jointly optimize the coding functions and the reconstruction network. Given that our exposure encodable camera supports only binary coding functions, we enforce hardware implementability by applying constraints based on a double well function and first-order difference. Additionally, Fisher information is incorporated into the loss to improve reconstruction quality, while Mean Squared Error (MSE) is used as the objective to guide the final output. Here we give more details about these losses.

**Mean Squared Error Loss.** We employ MSE as the fidelity loss to supervise the predicted depth map, as defined in Eq. 8

$$\mathcal{L}_{MSE} = \sum_s \|d_{pre}(s) - d_{gt}(s)\|_2^2 .\tag{8}$$

**Fisher Guidance Loss.** The SNR is one of the key factors influencing the quality of ToF imaging. Inspired by [11], we introduce the fisher guidance loss to enhance the quality of our depth reconstruction, which can be summarized as Eq. 9

$$\mathcal{L}_{fisher} = -\sum_s \sum_{i=1}^K \left[ \frac{1}{2\sigma_i^4(s)} + \frac{1}{\sigma_i^2(s)} \right] \left[ \frac{\partial \mathbb{E}(I_i(s))}{\partial d} \right]^2 , \tag{9}$$

where $\mathbb{E}(I_i(s))$ is the expectation of $I_i(s)$ and $\sigma_i(s) = \sqrt{\mathbb{E}(I_i(s)) + \mathbb{E}(n_d) + \sigma_r^2}$.

**Double Well Function Loss.** To enable the optimization of binary coding functions within the differentiable physical model. We introduce the double well function from quantum mechanics [36], which is formulated in Eq. 10

$$f_{dw}(x) = 4(x - 0.5)^4 - 2(x - 0.5)^2 . \tag{10}$$

As shown in Fig. 4, this function has two valleys at $x = 0$ and $x = 1$, thereby encouraging the coding functions to converge toward binary states during the optimization process. Therefore, our double well function loss can be defined as Eq. 11

$$\mathcal{L}_{dw} = \sum_{i=1}^K \sum_{j=1}^M f_{dw}(D_i(t_j)) , \tag{11}$$

where $M$ is the sampling points on each coding function.

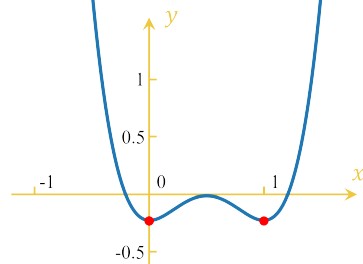

Figure 4: Demonstration of the double well function with two identical minima located at $x = 0$ and $x = 1$.

**First-order Difference Loss.** Although the double well function effectively constrains the coding functions to a binary state, we observe that the learned functions often exhibit extremely narrow peaks, which pose challenges for practical hardware implementation. To mitigate this issue, we introduce a first-order difference loss, as defined in Eq. 12. By minimizing the first-order difference loss, narrow peaks can be effectively suppressed, thus ensuring feasibility for hardware implementation.

$$\mathcal{L}_{1st} = \sum_{i=1}^K \sum_{j=1}^{M-1} |D_i(t_{j+1}) - D_i(t_j)| . \tag{12}$$

Finally, our complete loss can be summarized as Eq. 13

$$\mathcal{L} = \mathcal{L}_{MSE} + \gamma_1 \mathcal{L}_{fisher} + \gamma_2 \mathcal{L}_{dw} + \gamma_3 \mathcal{L}_{1st} , \tag{13}$$

where $\gamma_1$, $\gamma_2$ and $\gamma_3$ are loss balance coefficients.

## 4 Synthetic Assessment

### 4.1 Implementation Details

**Dataset.** We use the NYU-V2 dataset [37] to train and test our end-to-end framework. The NYU-V2 dataset is a high-quality RGB-D dataset captured by Kinect. It contains a total of 1449 pairs of precisely aligned RGB and depth images collected from 464 indoor scenes, which enables its extensive application in academic research. For each RGB-D pair, we first apply intrinsic image decomposition [38] to the RGB image to obtain reflectance and ambient light maps. Subsequently, as detailed in Sec. 3.1, given the reflectance $\rho_s$, ambient light $I_{amb}$, and depth $d(s)$, we can synthesize multiple BE-ToF measurements. We divide the dataset in detail, using 1000 pairs of data as the training set and the remaining 449 pairs as the test set [39, 40].

**Incremental Training Method.** In our BE-ToF system, the SNR varies not only with distance but also significantly under the same distance due to ambient light $I_{amb}$. Therefore, we introduce an incremental training strategy [41] to ensure robust depth estimation of our network under varying SNR levels. Specifically, for each distance, we define three distinct SNR scenarios arranged from high to low. The network is trained with input data of varying SNRs, progressively transitioning from high to low every 10 epochs. When data of all SNRs are traversed, samples with random SNR are generated and fed to the network for the convergence of the network.

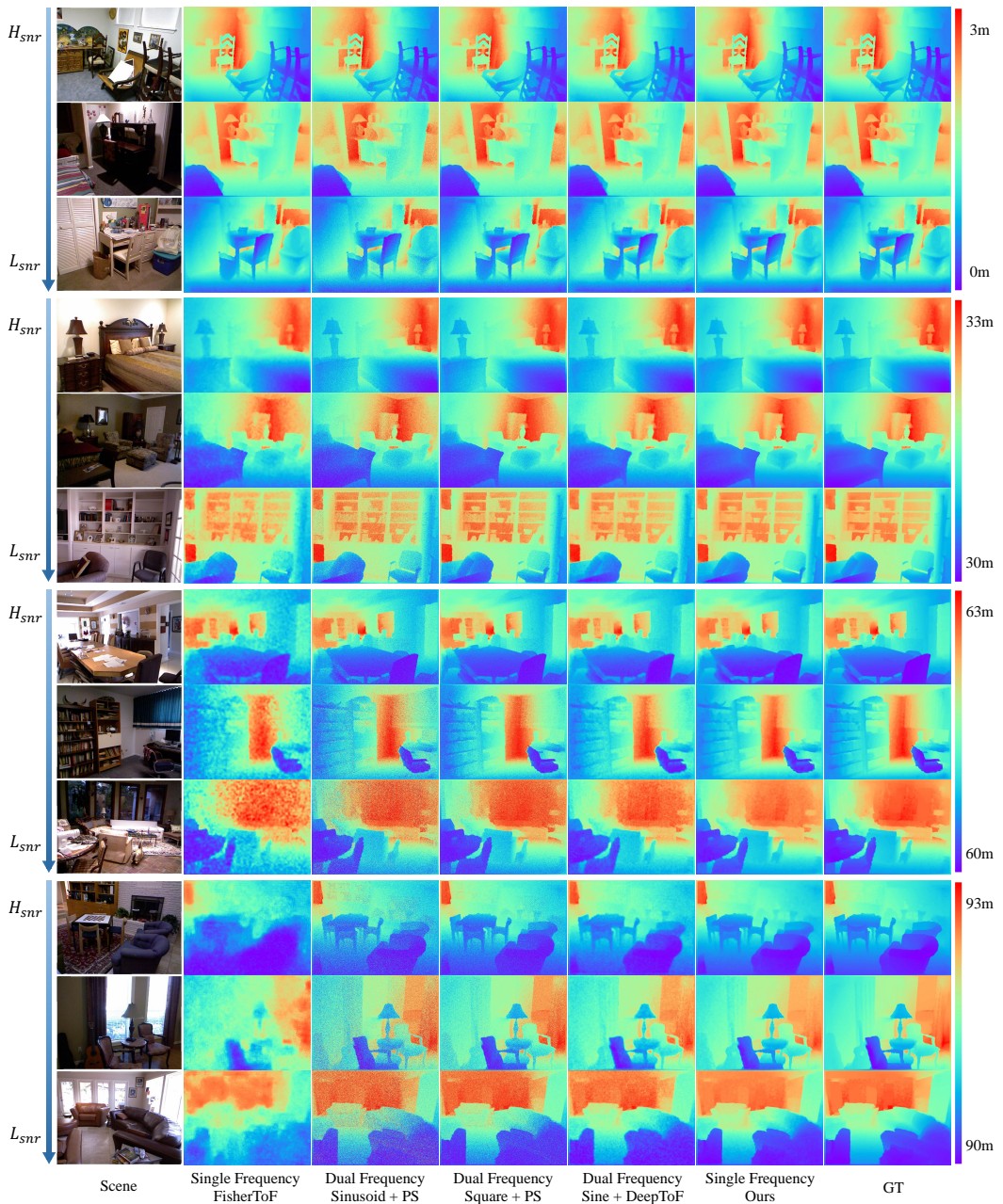

Figure 5: Overall comparisons with traditional iToF methods under various distances and SNRs, including FisherToF [11] under single frequency modulation; Sine/Square + PS algorithm [5] and Sine + DeepToF [23] under dual frequency modulation.

**Training Parameters.** We choose $K$ in Eq. 6 as 4 and $M$ in Eq. 11 as 1000. The number of restormer blocks in the network is set to $[4, 6, 6, 8]$. We train the network for 200 epochs using the ADAM optimizer [42] with a batch size of 20. The learning rate is initialized at 0.01 and decays by a factor of 0.7 every 10 epochs. The loss balance coefficients $\gamma_1$ and $\gamma_2$ are empirically set to 5e-4 and 5e-2 initially, and are updated to 5e-5 and 1 after 40 epochs. $\gamma_3$ is always set to 5. Xavier initialization is used for the learnable coding functions. All experiments are conducted on the PyTorch platform [43], using an NVIDIA GeForce RTX 4090 GPU.

Table 1: Quantitative comparison of overall performance, coding schemes, and reconstruction networks. All metrics are reported as MAE (mm).

| | 0-3m | | | 30-33m | | | 60-63m | | | 90-93m | | |
|---|---|---|---|---|---|---|---|---|---|---|---|---|
| | $H_{snr}$ | $M_{snr}$ | $L_{snr}$ | $H_{snr}$ | $M_{snr}$ | $L_{snr}$ | $H_{snr}$ | $M_{snr}$ | $L_{snr}$ | $H_{snr}$ | $M_{snr}$ | $L_{snr}$ |
| **(a) Overall Performance** | | | | | | | | | | | | |
| FisherToF [11] | 7.19 | 10.46 | 16.91 | 20.80 | 24.54 | 31.61 | 34.58 | 42.43 | 54.73 | 75.19 | 77.68 | 138.11 |
| Sine+PS [5] | 43.26 | 58.08 | 78.09 | 56.96 | 77.89 | 107.25 | 79.20 | 111.49 | 158.40 | 117.21 | 170.76 | 244.29 |
| Square+PS [5] | 33.21 | 40.64 | 51.29 | 40.06 | 51.19 | 66.90 | 51.90 | 69.12 | 93.62 | 72.16 | 100.12 | 140.98 |
| DeepToF [23] | 17.76 | 19.19 | 29.51 | 26.54 | 29.25 | 37.49 | 31.50 | 35.02 | 45.54 | 42.64 | 45.12 | 56.97 |
| **(b) Coding Scheme** | | | | | | | | | | | | |
| Square | 12.66 | 15.18 | 21.35 | 16.82 | 22.10 | 29.05 | 20.85 | 24.54 | 32.77 | 25.51 | 30.08 | 44.47 |
| **(c) Reconstruction Network** | | | | | | | | | | | | |
| DeepToF [23] | 14.76 | 16.20 | 20.32 | 18.25 | 23.30 | 34.12 | 24.95 | 31.50 | 37.90 | 28.12 | 38.17 | 48.55 |
| MaskToF [31] | 11.73 | 12.89 | 17.41 | 14.72 | 19.44 | 26.73 | 15.94 | 21.38 | 33.55 | 25.71 | 27.86 | 38.60 |
| FisherToF [11] | 6.94 | 8.10 | 16.26 | 10.22 | 15.71 | 21.68 | 14.62 | 18.31 | 29.73 | 22.60 | 24.89 | 31.83 |
| **Ours** | **5.90** | **6.95** | **12.71** | **8.03** | **12.25** | **18.29** | **11.93** | **16.60** | **26.08** | **18.96** | **21.99** | **29.58** |

## 4.2 Comparison with the State-of-the-art Methods

To demonstrate the superiority of our method, we conduct a detailed comparison with traditional iToF approaches, including single frequency modulation and dual frequency modulation. The scenarios encompass multiple distance ranges (0-3m, 30-33m, 60-63m, and 90-93m) combined with varying SNRs, specifically high ($H_{snr}$ = 22 dB), medium($M_{snr}$ = 19 dB) and low ($L_{snr}$ = 16 dB). As shown in Fig. 5, we first compare our method with FisherToF [11] under single frequency modulation. While FisherToF achieves precise depth reconstruction at close range, it still suffers from the rapid decline in imaging quality over distance. We then compare our method with a variety of dual frequency modulation approaches, including sinusoid and square coding functions with Phase Shift (PS) algorithm [5] and the learning-based DeepToF [23] method. Our method achieves the best performance across various distances and SNRs, using only single frequency modulation. We present a detailed quantitative comparison in Tab. 1 (a), with Mean Absolute Error (MAE) as the evaluation criterion.

We further substantiate the superiority of our method through an analysis of the learnable coding function and the proposed RSCF-Net. As for the coding function, considering the practical hardware implementability, we compare our learnable coding functions with the square coding function with the same RSCF-Net. The quantitative results presented in Tab. 1 (b) prove that our learnable coding functions provides superior depth reconstruction and enhanced robustness to noise. Additionally,

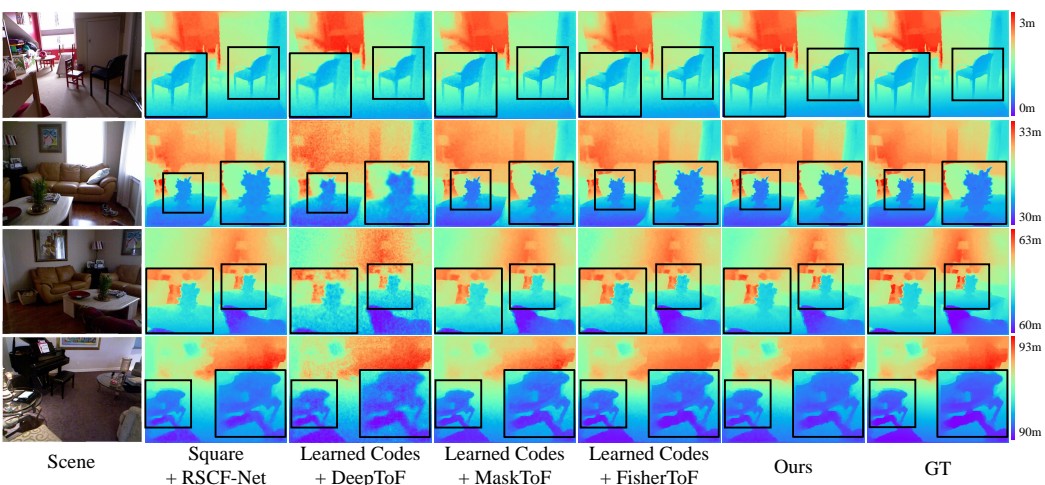

Figure 6: Comparisons with different coding scheme and reconstruction networks.

we perform a thorough comparison of our RSCF-Net with existing depth reconstruction networks with the same learned coding functions, including DeepToF [23], MaskToF [31] and FisherToF [11]. The quantitative results in Tab. 1 (c) confirm the effectiveness of our network. Fig. 6 presents the visual results of different methods across four distances under low SNR conditions, intuitively demonstrating the advantages of our approach.

### 4.3 Ablation Study

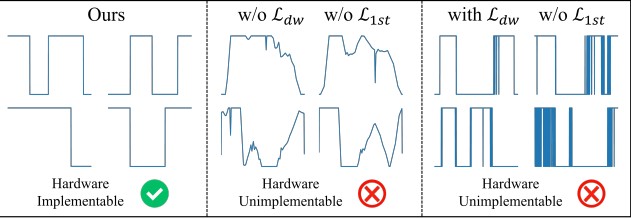

Figure 7: Visual ablations on $\mathcal{L}_{dw}$ and $\mathcal{L}_{1st}$.

Table 2: Quantitative ablations with MAE(mm) as the evaluation metric.

| Distance(m) | 0-3 | 30-33 | 60-63 | 90-93 |
|---|---|---|---|---|
| w/o $\mathcal{L}_{fisher}$ | 15.36 | 19.41 | 27.02 | 37.45 |
| w/o CFEB | 58.69 | 73.64 | 78.56 | 86.94 |
| w/o MFFB | 9.8 | 15.77 | 23.36 | 28.40 |
| w/o ECA | 10.22 | 13.01 | 21.41 | 26.06 |
| **Ours** | **8.52** | **12.86** | **18.20** | **23.51** |

We first perform ablations on the proposed double well function loss $\mathcal{L}_{dw}$ and first-order difference loss $\mathcal{L}_{1st}$. Since the learned coding functions are primarily used to control the camera's exposure, they must be strictly binary. As illustrated in Fig. 7, the absence of $\mathcal{L}_{dw}$ and $\mathcal{L}_{1st}$ results in coding functions that are entirely impractical to implement in hardware. With only the $\mathcal{L}_{dw}$, the coding functions do converge to binary states; however, the proliferation of narrow peaks still makes them impossible to implement on real hardware.

In the next, we present a quantitative analysis to evaluate the impact of the fisher guidance loss and different network blocks. The values in Tab. 2 represent the average MAE measured under different SNRs at the same distance. The experimental results demonstrate the effectiveness of the introduced Fisher loss in guiding the network to learn an optimal coding functions. The ablation studies on different network blocks further validate the significant improvement in reconstruction quality brought by the proposed CFEB and MFFB.

## 5 Physical Experiment Results

**Hardware Prototype.** As shown in Fig. 8, to validate the effectiveness of our BE-ToF approach in real world scenarios, we built a prototype system comprising a solid-state pulsed laser and an exposure-encodable ICMOS sensor. The laser operates at 532 nm with a 5 ns pulse width, a fixed 1 kHz repetition rate, and up to 1 mJ single-pulse energy. To realize area illumination, we homogenize the beam with a diffuser and expand it using a beam expander. The ICMOS is fitted with a zoom lens (300-800 mm) and supports a minimum exposure gate of 3 ns. Timing synchronization is provided by a fast photodiode that detects each laser pulse and issues a hardware trigger to a signal generator, which then drives the ICMOS with the learned coding functions. This hardware chain achieves picosecond-scale synchronization, ensuring high-quality imaging.

**Experimental Results.** As shown in Fig. 9, we evaluate our approach across diverse indoor and outdoor scenarios, including a hand model and a kettle indoors and a stone model and a stair out-

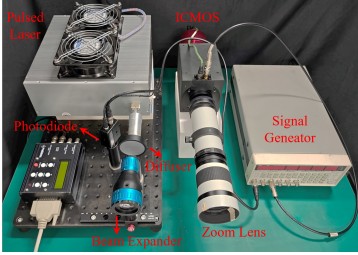

Figure 8: System Prototype.

Table 3: Quantitative results on real world experiments with MAE(cm) as the evaluation metric.

| | Square + PS | Square + RSCF-Net | Learned Codes + DeepToF | Learned Codes + FisherToF | Ours |
|---|---|---|---|---|---|
| Hand | 10.57 | 3.96 | 4.73 | 2.47 | 1.78 |
| Kettle | 9.13 | 3.51 | 4.14 | 1.99 | 1.41 |
| Stone | 12.16 | 4.76 | 5.94 | 3.78 | 2.50 |
| Stair | 15.79 | 5.92 | 6.70 | 4.26 | 3.83 |

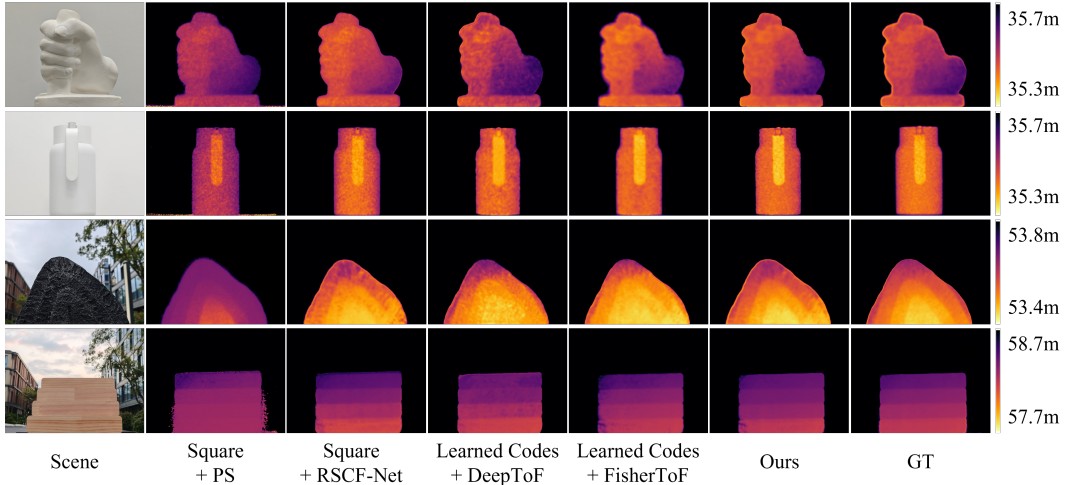

Figure 9: Real world experiments across indoor and outdoor scenarios.

doors. All experiments use the same settings as in simulation: we apply the coding functions learned in simulation and reconstruct with RSCF-Net. The modulation period $T_m$ is fixed at 20 ns, and the burst period $T_{burst}$ is set to 1ms, corresponding to the laser repetition rate of 1 kHz. We perform a detailed comparison against other methods, including square coding function and several reconstruction networks. Quantitative results are summarized in Tab. 3. Ground truth is acquired via a time-delay scan at the minimum exposure time (3 ns) with a 1 ns step. Both qualitative and quantitative results demonstrate that our system consistently achieves centimeter-level depth accuracy across these scenarios and outperforms existing methods.

## 6  Conclusion, Limitations, and Broader Impact

In conclusion, we propose a novel ToF imaging paradigm, termed BE-ToF. The BE-ToF system enables long-distance high-fidelity depth imaging by modulating and demodulating pulsed signals in burst mode using only single-frequency modulation. Additionally, we introduce a learnable end-to-end framework that jointly optimizes binary coding functions and the reconstruction network to effectively handle varying SNRs across different distances, achieving state-of-the-art performance.

**Limitations.**   Despite achieving both long-distance and high-fidelity depth imaging, our BE-ToF system is subject to limitations in its imaging range. As shown in Fig. 1, the operational range is confined between $\frac{c \cdot \tau}{2}$ and $\frac{c \cdot (\tau + T_m)}{2}$, with higher precision resulting in a narrower imaging range. We are currently exploring several promising directions to mitigate these limitations. First, we can exploit BE-ToFs flexible time-delay control to perform temporal scanning and synthesize a wide-range depth map. Second, because temporal scanning can incur significant latency, we favor a coarse-to-fine strategy: first capture a wide-range, low-resolution depth map, then use BE-ToF to selectively acquire high-precision depth in regions of interest (ROIs).

**Broader Impact.**   The proposed BE-ToF system demonstrates strong potential for applications such as autonomous driving and topographic surveying, offering enhanced reconstruction quality and improved processing efficiency. However, its ability to perform long-distance depth imaging raises potential privacy concerns, particularly in scenarios where individuals may be unknowingly captured. Addressing these concerns responsibly is essential for real-world deployment.

## Acknowledgments and Disclosure of Funding

This work was supported by the National Key Research and Development Program of China No. 2022YFA1207200, National Natural Science Foundation of China No. 62522113, 62505132, the Fundamental Research Funds for Central Universities No. 021014380227, 021014380260, and Open Research Project of Suzhou Laboratory No. SZLAB-1508-2024-TS015.

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

# A Technical Appendices and Supplementary Material

## A.1 Comparison with Other Coding Functions

To fully highlight the advantages of our learnable binary coding functions, we conduct a detailed comparison against alternative coding functions, including sinusoid, square, Hamiltonian [44], and M-sequence [45]. To ensure a fair comparison of coding functions, we employ the same imaging setup (with K = 4 measurements) and reconstruction network, varying only the coding functions. The quantitative results, evaluated using Mean Absolute Error (MAE) in millimeters, are summarized in Tab. 4. It can be observed that, under the same reconstruction network, our coding functions delivers the best performance. Notably, although we compare multiple coding functions in simulation, only the square coding function and our learnable binary coding functions are implementable on actual hardware. The sinusoidal coding function is excluded due to its non-binary nature, while Hamiltonian codes and M-sequences contain narrow peaks that are impractical to implement given hardware constraints on minimum exposure time.

To further demonstrate the advantages of optimizing coding functions using neural networks, we use Fisher information to evaluate the quality of each coding function. As discussed in [11], Fisher information can be used as a metric to assess the optimality of different coding schemes a higher Fisher Information value indicates a more optimal coding scheme. Therefore, we list the Fisher Information of different coding functions in Tab. 5 for a straightforward comparison. It can be observed that our learnable binary coding functions achieves the highest Fisher information, which demonstrates its optimality.

Table 4: Quantitative comparison with other coding functions.

|  | 0-3m | 30-33m | 60-63m | 90-93m |
|---|---|---|---|---|
| Sinusoid | 24.67 | 31.12 | 39.80 | 45.39 |
| Square | 16.40 | 22.66 | 26.05 | 33.35 |
| Hamiltonian [44] | 11.28 | 14.53 | 21.74 | 27.11 |
| M-sequence [45] | 15.19 | 21.24 | 28.33 | 38.81 |
| **Ours** | **8.52** | **12.86** | **18.20** | **23.51** |

Table 5: Quantitative comparison of Fisher Information with other coding functions.

|  | Sinusoid | Square | Hamiltonian | M-sequence | **Ours** |
|---|---|---|---|---|---|
| Fisher Information | $1.27 \times 10^6$ | $2.29 \times 10^6$ | $2.92 \times 10^6$ | $2.18 \times 10^6$ | $\mathbf{3.93 \times 10^6}$ |

## A.2 Experiments with Fewer Measurements

As discussed in Sec. 3.1, at least $K \geq 3$ measurements are required to recover depth. In our work, we choose $K = 4$ to ensure high reconstruction quality and robustness to noise. In Tab. 6, we present quantitative results for cases with $K \leq 4$, evaluated using Mean Absolute Error (MAE) in millimeters. When $K < 3$, the reconstruction quality significantly degrades, which is reasonable given the limited information available for depth recovery. With $K = 3$, depth can be reasonably reconstructed, though still slightly inferior to $K = 4$, where the additional measurement improves robustness to noise and other perturbations.

Table 6: Quantitative comparison with different measurements.

|  | 0-3m | 30-33m | 60-63m | 90-93m |
|---|---|---|---|---|
| $K = 1$ | 133.22 | 166.90 | 156.18 | 198.19 |
| $K = 2$ | 37.84 | 42.94 | 51.96 | 49.18 |
| $K = 3$ | 14.58 | 17.59 | 21.64 | 29.88 |
| $K = 4$ | 8.52 | 12.86 | 18.20 | 23.51 |

## A.3 Ablations on Loss Balance Coefficients

In the loss defined in Eq. 13, the MSE term is the principal objective driving accurate depth reconstruction, while the three additional regularization terms serve as auxiliary constraints that guide the learning of the binary coding functions. To validate our choice of loss weights, we conduct a comprehensive ablation over the coefficients and report the results. All experiments are performed at distances of 0-3 m.

For coefficient $\gamma_1$ before 40 epochs, as shown in Tab. 7, setting $\gamma_1 < 5 \times 10^{-6}$ makes its effect too weak for the network to learn effective coding functions, leading to performance drop. When it exceeds $5 \times 10^{-4}$, it disrupts the double-well and first-order difference losses, resulting in coding functions unimplementable for hardware. Thus, we set $\gamma_1$ to $5 \times 10^{-4}$ during the first 40 epochs.

Table 7: Ablations on $\gamma_1$ before 40 epochs.

| $\gamma_1$ Before 40 Epochs | $5 \times 10^{-7}$ | $5 \times 10^{-6}$ | $5 \times 10^{-5}$ | $5 \times 10^{-4}$ | $5 \times 10^{-3}$ | $5 \times 10^{-2}$ |
|---|---|---|---|---|---|---|
| MAE(mm) | 17.23 | 12.47 | 9.76 | 8.52 | Hardware Unimplementable | Hardware Unimplementable |

For coefficient $\gamma_1$ in epochs after 40, as shown in Tab. 8, we find that the value of $\gamma_1$ has little impact on the final performance. However, setting it too high can slow down the convergence of the network. Therefore, we set $\gamma_1$ to 5e-5 after 40 epochs to balance performance and convergence speed.

Table 8: Ablations on $\gamma_1$ after 40 epochs.

| $\gamma_1$ After 40 Epochs | $5 \times 10^{-7}$ | $5 \times 10^{-6}$ | $5 \times 10^{-5}$ | $5 \times 10^{-4}$ | $5 \times 10^{-3}$ | $5 \times 10^{-2}$ |
|---|---|---|---|---|---|---|
| MAE(mm) | 9.94 | 8.73 | 8.52 | 14.58 | 11.62 | 10.17 |
| Convergence Epochs | 107 | 103 | 112 | 123 | 133 | 137 |

For coefficient $\gamma_2$ in epochs before 40, we set $\gamma_2$ to a small value so that the first-order difference loss dominates and helps suppress narrow peaks. As shown in Tab. 9, when $\gamma_2$ exceeds 1, the learned coding functions exhibit narrow peaks and become unsuitable for hardware implementation. Thus, we set $\gamma_2$ to $5 \times 10^{-2}$ during the first 40 epochs of training.

Table 9: Ablations on $\gamma_2$ before 40 epochs.

| $\gamma_2$ Before 40 Epochs | $5 \times 10^{-4}$ | $5 \times 10^{-3}$ | $5 \times 10^{-2}$ | $5 \times 10^{-1}$ | 1 | 5 |
|---|---|---|---|---|---|---|
| MAE(mm) | 11.98 | 12.35 | 8.52 | 12.67 | 10.27 | Hardware Unimplementable |

For coefficient $\gamma_2$ in epochs after 40, we increase the value of $\gamma_2$ to encourage the coding functions to converge more rapidly to a binary state. As shwon in Tab. 10, setting the coefficient below $5 \times 10^{-2}$ prevents the coding functions from reaching a binary state, while values above 30 cause noticeable performance degradation. Therefore, we set $\gamma_2$ to 1 after 40 epochs.

Table 10: Ablations on $\gamma_2$ after 40 epochs.

| $\gamma_2$ After 40 Epochs | $5 \times 10^{-3}$ | $5 \times 10^{-2}$ | $5 \times 10^{-1}$ | 1 | 12 | 20 | 30 | 40 |
|---|---|---|---|---|---|---|---|---|
| MAE(mm) | Hardware Unimplementable | 10.26 | 9.02 | 8.52 | 9.29 | 11.41 | 10.68 | 17.92 |

For coefficient $\gamma_3$, as shown in Tab. 11, when the coefficient is too small, narrow peaks appear, making hardware unimplementable. Conversely, a large coefficient results in degraded depth recon-

struction quality. A balanced performance is achieved with values between 0.05 and 10; we set it to 5 in our experiments.

Table 11: Ablations on $\gamma_3$.

| $\gamma_3$ | $5 \times 10^{-4}$ | $5 \times 10^{-3}$ | $5 \times 10^{-2}$ | 1 | 5 | 10 | 20 | 30 |
|---|---|---|---|---|---|---|---|---|
| MAE(mm) | Hardware Unimplementable | Hardware Unimplementable | 9.02 | 8.98 | 8.52 | 8.53 | 16.88 | 23.76 |

### A.4 Simulation Method for the Long-Range Indoor Dataset

We use the NYU Depth V2 RGB-D dataset to train and evaluate our network. First, we scale the depth values to a fixed maximum range of 3 m (corresponding to $T_m$=20ns), and apply this setting consistently across all simulation experiments for fairness. As discussed in Sec. 3.1, ToF imaging is principally affected by ambient light, scene reflectance, and depth. To model these factors, we perform intrinsic image decomposition [38] on the RGB images to separate ambient light and reflectance components, and we calibrate a distance-dependent attenuation curve under long-range conditions within the simulation environment. Because ambient illumination and scene reflectance are depth-invariant in this model, variations across distance are primarily attributed to attenuation. Accordingly, given a fixed emitted signal, we apply distance-dependent attenuation coefficients to the reflected signal and then synthesize the corresponding ToF measurements. In addition, we explicitly model sensor noise as in Eq. 7, incorporating both dark current and readout noise to obtain more realistic measurements.

It is worth noting that our setting still differs from real long-range outdoor scenarios in several respects: (i) the current simulation does not model atmospheric effects, which can significantly influence ToF imaging outdoors; and (ii) beam divergence remains a key factor affecting image quality. To mitigate the latter, we employ a laser beam expander to generate area illumination, though some deviation from ideal uniform lighting persists. In future work, we plan to incorporate atmospheric effects into the simulation framework to better align with real-world experiments; meanwhile, spatial filtering is used to further improve illumination uniformity.

### A.5 Precision of Binary Coding Function Optimization

In this work, we adopt a differentiable double well function to drive the coding functions toward 0 or 1, thus ensuring that our end-to-end framework is fully differentiable. While the optimized coding functions are effectively binary, they do not attain exact binary values in floating-point arithmetic. Empirically, the learned coding functions lie extremely close to the binary extremes (e.g., around 0.0001 or 0.999), and we regard such deviations as negligible for our network. To validate our conclusion, we apply a round function during testing to convert coding functions into strict binary states(0 or 1) and compare results without it. The quantitative results in Tab. 12 show that strict binarization does not cause performance degradation; on the contrary, it slightly improves performance.

Table 12: Quantitative comparison of coding functions with/without round function with MAE(mm) as the evaluation metric.

| | 0-3m | 30-33m | 60-63m | 90-93m |
|---|---|---|---|---|
| Without Round Function | 8.52 | 12.86 | 18.20 | 23.51 |
| With Round Function | 8.51 | 12.78 | 17.97 | 21.97 |

### A.6 Additional Results on Other Datasets

To rigorously assess the generalization capability of our method, we conduct experiments on the 4D Light Field [46] and SUN RGB-D [47] datasets, selecting 16 scenes from the former and 298 scenes from the latter as the test sets. Both datasets are processed according to the approach detailed in the main manuscript: each RGB-D pair undergoes intrinsic image decomposition to separate it into an albedo map and a shading map. Specifically, the R-channel of the albedo map is utilized as the

albedo component, while the average of its three RGB channels serves as the ambient illumination. The network, trained exclusively on the NYU-V2 [37] dataset, is then evaluated on these datasets without any fine-tuning. For comparison, we include several baseline methods: FisherToF [11] under single frequency modulation, and sinusoidal and square coding functions combined with the Phase Shift (PS) algorithm [5], as well as the learning-based DeepToF [23], all under dual-frequency modulation. The quantitative results are summarized in Tab. 13. As shown, our method exhibits strong generalization performance across both datasets and achieves the highest reconstruction accuracy among all compared approaches. Additionally, we present qualitative results on the 4D Light Field and SUN RGB-D datasets in Fig. 10 and Fig. 11, respectively. The visualizations further confirm that our method delivers robust reconstruction performance, even under challenging conditions such as long-range scenes or low signal-to-noise ratios.

Table 13: Quantitative comparison of overall performance on 4D Light Field [46] and SUN RGB-D [47] dataset.

| | 0-3m | | | 30-33m | | | 60-63m | | | 90-93m | | |
|---|---|---|---|---|---|---|---|---|---|---|---|---|
| | $H_{snr}$ | $M_{snr}$ | $L_{snr}$ | $H_{snr}$ | $M_{snr}$ | $L_{snr}$ | $H_{snr}$ | $M_{snr}$ | $L_{snr}$ | $H_{snr}$ | $M_{snr}$ | $L_{snr}$ |
| **(a) Overall Performance on 4D Light Field Dataset** | | | | | | | | | | | | |
| Sine+PS [5] | 43.39 | 58.20 | 78.10 | 57.07 | 78.01 | 106.93 | 79.37 | 111.08 | 157.91 | 116.95 | 170.50 | 247.06 |
| Square+PS [5] | 33.15 | 40.58 | 51.20 | 39.99 | 51.11 | 66.80 | 51.83 | 69.04 | 93.32 | 72.05 | 99.72 | 140.14 |
| DeepToF [23] | 19.16 | 23.27 | 40.73 | 34.10 | 36.91 | 45.53 | 38.48 | 42.08 | 52.98 | 54.44 | 55.25 | 66.11 |
| FisherToF [11] | 8.83 | 12.81 | **19.19** | 27.19 | 31.99 | 39.04 | 44.72 | 52.95 | 65.41 | 83.79 | 98.05 | 140.15 |
| **Ours** | **6.79** | **11.06** | 19.64 | **11.35** | **16.29** | **26.75** | **15.11** | **21.13** | **32.25** | **26.78** | **28.78** | **34.17** |
| **(b) Overall Performance on SUN RGB-D Dataset** | | | | | | | | | | | | |
| Sine+PS [5] | 40.46 | 54.10 | 72.39 | 53.06 | 72.20 | 98.64 | 73.39 | 102.39 | 144.44 | 107.51 | 155.64 | 224.36 |
| Square+PS [5] | 31.90 | 38059 | 48.25 | 38.06 | 48.15 | 62.44 | 48.79 | 64.46 | 86.58 | 67.20 | 92.42 | 129.06 |
| DeepToF [23] | 16.47 | 23.45 | 30.71 | 25.12 | 32.67 | 36.78 | 27.91 | 35.80 | 40.99 | 36.07 | 45.11 | 55.05 |
| FisherToF [11] | 8.08 | 11.80 | 18.96 | 21.35 | 25.39 | 31.83 | 33.34 | 40.62 | 51.86 | 74.61 | 76.92 | 132.77 |
| **Ours** | **6.63** | **7.53** | **12.11** | **9.00** | **12.55** | **18.03** | **11.59** | **16.85** | **29.20** | **17.93** | **21.62** | **31.70** |

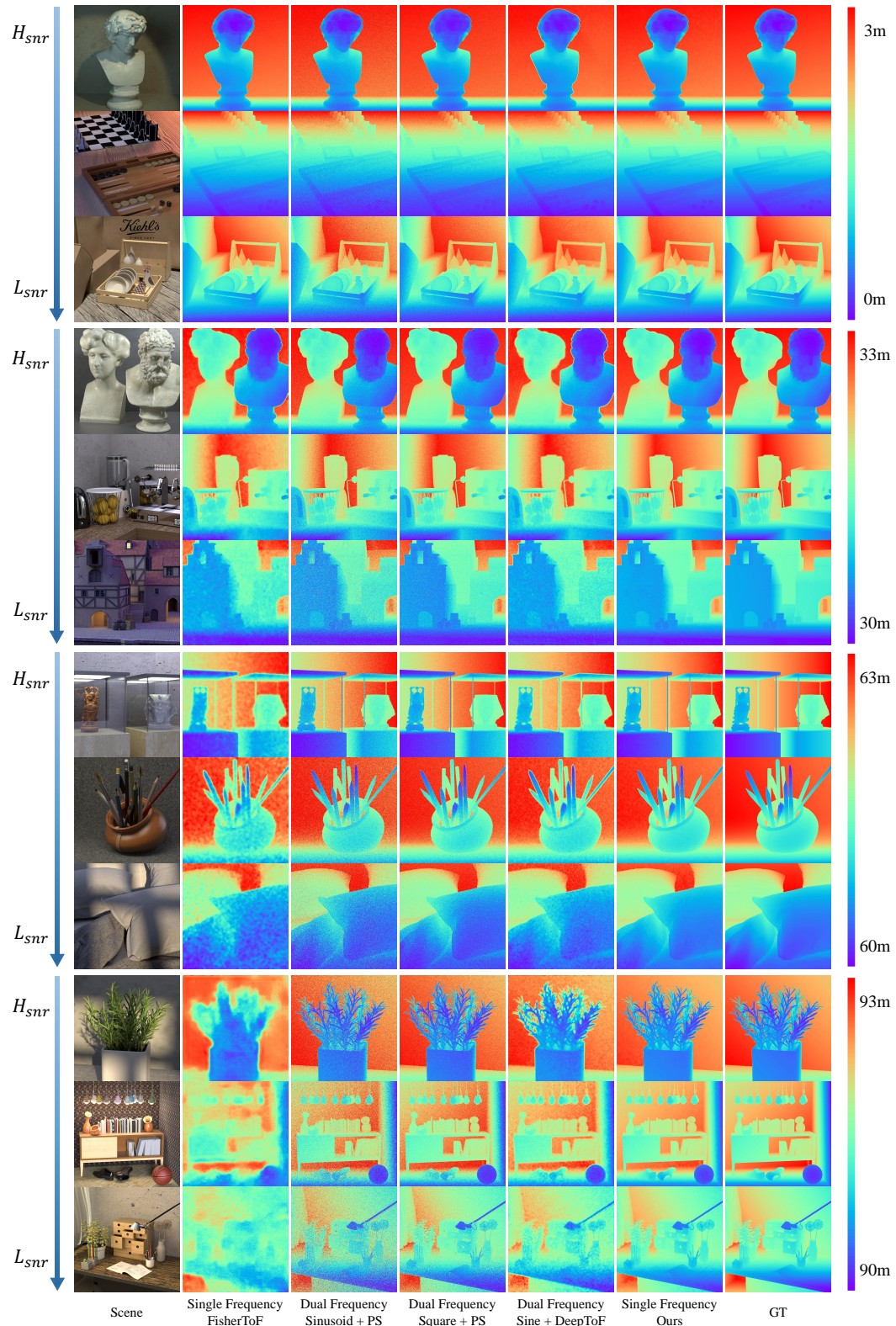

Figure 10: Overall comparisons with traditional iToF methods under various distances and SNRs on 4D Light Field dataset, including FisherToF [11] under single frequency modulation; Sine/Square + PS algorithm [5] and Sine + DeepToF [23] under dual frequency modulation.

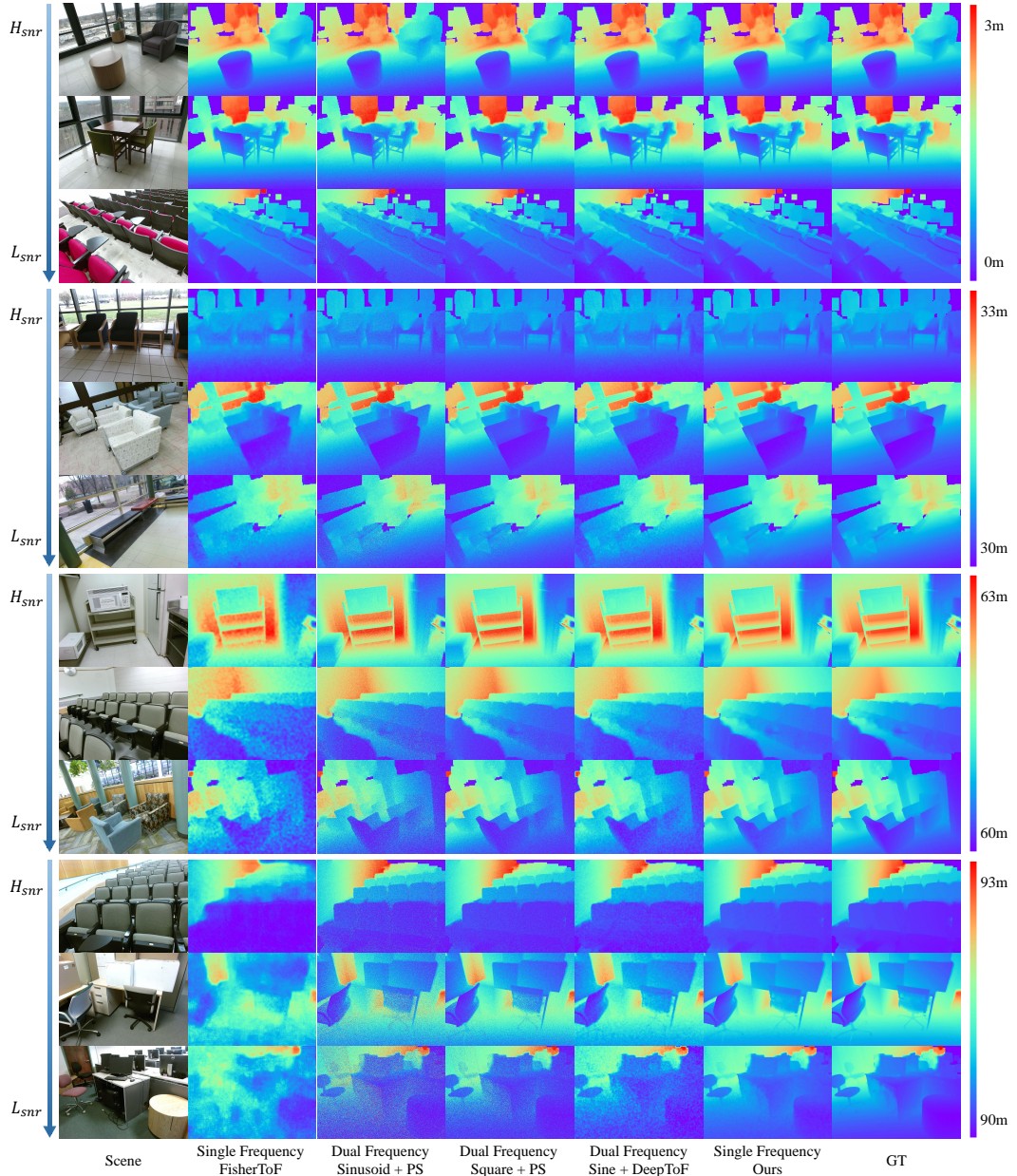

Figure 11: Overall comparisons with traditional iToF methods under various distances and SNRs on SUN RGB-D dataset, including FisherToF [11] under single frequency modulation; Sine/Square + PS algorithm [5] and Sine + DeepToF [23] under dual frequency modulation.

