# OpenReview forum: "Learnable Burst-Encodable Time-of-Flight Imaging for High-Fidelity Long-Distance Depth Sensing"
_NeurIPS.cc/2025/Conference — NeurIPS 2025 spotlight_

### Official Review · Reviewer_56Ri · 2025-06-23

**Clarity:** 2
**Significance:** 4
**Originality:** 3
**Rating:** 5
**Confidence:** 4

**Summary:**

This paper proposes Burst-Encodable Time-of-Flight (BE-ToF) and an end-to-end learnable reconstruction framework that jointly optimizes the coding function and ToF signals to obtain a better quality depth map.

BE-ToF uses a short-period Modulation signal to have the advantage of using a high-frequency modulation signal (e.g. better details in the long distance) while adding a time delay in the demodulation signal to get over a phase wrapping issue on the far distance region.

The reconstruction network is based on RSCF-Net and takes synthetically generated raw ToF images using the BE-ToF Imaging model. This network can optimize both learnable coding functions that can optimize the demodulation signal of the ToF, and the network itself, which denoises and improves the ToF signals to produce high-quality depth in the end.

In the end, the paper shows the prototype of BE-ToF depthmap that uses the learned coding function and BE-ToF principle.

**Questions:**

1. Lack of Clarity

- Somehow, the explanation regarding the advantage of BE-ToF in Sec.3 is not really clear. Personally, what is written in the first and second paragraphs of the supplementary material helped me more to understand the purpose of BE-ToF. It would be great to merge the Suppl. material's BE-ToF principle to Sec.3

- How the differential BE-ToF Imaging Model is used is not well written in the paper. It would be great to refer to Fig.2 in Section 3.1, then the use of the model will be clearer.

- Also, It would be great to mention in the dataset paragraph  (4.1)  that the raw ToF images are generated/synthesized from Sec. 3.1, Eq.5. It was not really clear how the train is done from the conventional depth dataset but with the input of raw ToF signal in the first iteration of reading. Mentioning that briefly will make readers understand the paper much more easily.

- It would be great to mention the reasoning behind the selection of RSCF-Net briefly.

- The depth range in Fig.5 is created by rescaling the GT depth? Also, I see the range is always 3m. Is it because the short-period signal that goes up to 3m is used for the experiment? Then it would be great to mention briefly in Sec.4.2.

- It's hard to understand lines 189-190 regarding albedo. In Eq2, roh_s is reflectance (line 123), but in lines 189-190 it is mentioned as albedo. Also, it's hard to understand why using the R channel of albedo is used as roh_s

2. BE-ToF's limitation of being able to acquire only in a specific range of depth maps

- This is more of a limitation of BE-ToF functionality. It's great to have a ToF that can capture the far distance without having issues with phase wrapping. But having a small distance window limits its real-life use case. It would be great to have some way to improve this issue or have a discussion on this as future work.

3. Limited Originality

- This paper mainly shows details about the reconstruction network and relatively small details about BE-ToF, but the network part seems to be an incremental work from [10] (in the main paper), which changed the reconstruction network backbone and added more loss in the forward model, and the network is not necessarily tied to the BE-ToF modality as well.
- As proposing BE-ToF is the main differentiation from [10] (in the main paper), i'd love to see experiments that run more with real-life BE-ToF instead of synthetic training. I posted more comments regarding the experiment in the next section.

4. Limited real ToF dataset experiment

- It's understandable that doing a real-life experiment is challenging in this setup. But the experiments are mostly synthetic, and only one picture of a real-life ToF image is included. In a real-life scenario, raw ToF images are really noisy, sparse, having physically induced noise like MPI, Motion Blur, and artifacts from surface material of the objects in the scene, which can make the input of the raw ToF signal significantly different from the simple ToF model introduced in the paper.
- Also, experiment on Sec.5 doesn't seem to use what was trained from the network training (at least its not mentioned), if the network is not used in real, the readers can understand that the real raw signal is not really close to what is used in the synthetic trained input. Synthetic-based trained ToF models still show results on the real ToF setup, for example, from DeepToF or Gu et al [1]. If the depth in Fig.8 is from the BE-ToF and Network, then please specify. And if not, it would be great to see how the result looks on the trained network.
- As far as I see, the reconstruction network itself is independent to how the raw ToF signal is acquired. One suggestion is to test the reconstruction network on the real-life ToF dataset that contains raw ToF and depth map as GT. This way, the authors can prove the performance of the proposed network compared to the existing ToF pipelines. As far as I checked, there are two real datasets with raw ToF included (CroMo dataset [2], HAMMER dataset [3])
- Another possible experiment is to modify the system prototype to make the signal compatible with other trained baselines (e.g. changing frequency and without delay to try FisherToF model, adding dual frequency model to test on DeepToF model) and show the qualitative experiment on the depth images. It would be great to see whether the real-life quality of depth on different setups acts similarly to what is shown in Fig.5.

Minor suggestion but, there is not much related work on ToF imaging. There are two more works that are quite relevant to this paper to be added in the related work: [1] (as multi-frequency modulation, and learnable kernel), [4] (RGB-based phase unwrapping from raw ToF)

[1] Tackling 3D ToF Artifacts Through Learning and the FLAT Dataset, Guo. et a.

[2] CroMo: Cross-Modal Learning for Monocular Depth Estimation, Verdié et.al.

[3] On the Importance of Accurate Geometry Data for Dense 3D Vision Tasks, Jung et.al.

[4] Wild ToFu: Improving Range and Quality of Indirect Time-of-Flight Depth with RGB Fusion in Challenging Environments, Jung et.al

**Ethical Concerns:**

["NO or VERY MINOR ethics concerns only"]

**Final Justification:**

The authors clarified the questions regarding the setup used in the paper and especially provided extra real-life experiments with quantiative evaluations, which were the major drawback of the paper. Given that the authors have polished the paper according to the rebuttal, the paper will be in a reasonable state.

**Limitations:**

Yes

**Quality:**

2

**Strengths And Weaknesses:**

Strength :
1. The paper proposes a fundamental solution to the existing issue with i-ToF sensing
2. The authors built a real-life prototype of the sensor that is proposed in the paper and shows it actually works in a real-life scenario
3. Both Qualitative and Quantiative experiments show superior quality of the proposed modality and the nework

Weakness (suggestions for each weakness are written in the Qestions part)
1. Lack of Clarity: The paper includes many details in different parts, but the connection is very weak between the details.
2. BE-ToF's limitation: BE-ToF is able to acquire only in a specific range of depth maps.
3. Limited Originality:  The paper's writing focuses a lot on the network and training part, while the network and training rather seem to be an incremental work from [10], which is rather chaing the backbone and adding more losses
4. Limited real ToF dataset experiment: Experiments are mainly done on the synthetic training while only one image is added for the real life scenario.

---

> ### Author Rebuttal · Authors · 2025-07-27
>
> We appreciate the reviewer’s recognition of the effectiveness of our method. In this rebuttal, we provide detailed responses to the main concerns raised in the review.
>
> > **Q1: Lack of clarity.**
>
> **A1:**
>
> 1. The key advantage of our BE-ToF system is its capability to achieve long-distance and high-precision depth imaging with only a single modulation frequency. In addition, as shown in Fig. 1(b), our depth accuracy and range depend solely on the period $T_m$. Therefore, our BE-ToF system is highly robust to the laser's pulse repetition frequency (PRF) and can operate reliably across a wide range (ranging from tens of Hz to hundreds of kHz), significantly reducing the hardware implementation complexity. We sincerely appreciate your valuable suggestion and will integrate the description of the BE-ToF principle in the camera-ready version.
>
> 2. As illustrated in Fig. 2(b), the differentiable forward model is designed to simulate ToF measurements from ground-truth depth. We first compute the reflected signal $R(s,t)$ in Eq. 3 based on the GT depth. Subsequently, the ToF measurements $I_i(s)$ in Eq. 4 are synthesized using the learnable binarized coding functions. Taking into account the noise characteristics of camera sensor, we introduce readout noise and dark noise to obtain more realistic measurements $X_i(s)$ in Eq. 5. We will carefully revise Sec. 3.1 and Fig. 2 to ensure better consistency and clarity.
>
> 3. It‘s true that the ToF measurements are simulated from the depth dataset following the method described in Sec. 3.1. We will clarify this point in Sec. 4.1 to ensure better readability.
>
> 4. We appreciate your suggestion. In RSCF-Net, we adopt Restormer as the backbone due to its strong ability to model long-range pixel interactions and  its lightweight architecture. Moreover, unlike conventional image restoration, ToF depth imaging involves reconstructing a depth map from multiple frames of ToF measurements, which places greater emphasis on capturing inter-channel dependencies. To address this, we design the CFEB and MFFB modules to more effectively fuse information across channels. As presented in Tab. 1 of the main manuscript, the proposed RSCF-Net achieves state-of-the-art performance. Furthermore, Tab. 1 in the supplementary material demonstrates its strong generalization capability across diverse datasets. We also conduct a systematic comparison of various networks regarding their computational complexity (FLOPs), parameter count. The detailed results are summarized below.
>
> |                    | DeepToF | MaskToF | FisherToF | RSCF-Net (Ours) |
> | :----------------: | :-----: | :-----: | :-------: | :-------------: |
> |   **FLOPs (G)**    |  12.65  |  5.62   |   26.72   |      14.69      |
> | **Parameters (M)** |  42.57  |  18.12  |   27.87   |      13.04      |
>
> 5. To ensure fair comparisons, all experiments in Sec. 4 are conducted with a fixed depth range of 3 meters (corresponding to $T_m$=20 ns). For the NYU-V2 dataset, we first scale the depth values to a 3-meter range. Then, to simulate the same depth range at different distances, we adjust the light attenuation coefficients accordingly. We will incorporate these clarifications into Sec. 4.2 as suggested.
>
> 6. We appreciate your careful reading. As you correctly noted, $\rho_s$ denotes reflectance as defined in line 123. We will revise the terminology in line 189 for consistency. For the definition of reflectance, we refer to the intrinsic image decomposition introduced in reference [35] of the manuscript. In [35], given $p$ is a pixel position, an image $I(p)=R(p) \cdot S(p)$ can be decomposed into shading layer $S(p)$ and reflectance layer $R(p)$. The shading layer $S(p)$ depicts the amount of reflected light, while the reflectance layer $R(p)$ depicts the intrinsic color of the material at $p$. Furthermore, $R(p)$ is decomposed as $R(p) = R_B(p) \cdot T(p)$, where $R_B$ is the base reflectance and $T$ represents texture details. Based on this, we perform intrinsic image decomposition on the NYU-V2 dataset and use the base reflectance layer $R_B$ as $\rho_s$ in Eq. 2.
>
>
>
>
>
>
>
> > **Q2: BE-ToF's limitation of being able to acquire only in a specific range of depth maps.**
>
> **A2:** We sincerely appreciate your interest in our work. While we acknowledge the current limitations of BE-ToF, we are actively investigating several promising directions to address them in future research. One practical approach leverages BE-ToF’s flexible control over the depth range via time-delay adjustment to perform temporal scanning, thereby synthesizing a comprehensive, wide-range depth map. Considering the high latency potentially introduced by temporal scanning, we propose leveraging compressive sensing to reduce the amount of data acquisition. In addition, a more ideal approach is to adopt a coarse-to-fine strategy: initially capturing a wide-range but low-resolution depth map, followed by selective high-precision depth acquisition in regions of interest (ROIs) using BE-ToF. This approach can be further enhanced with AI-driven techniques to enable intelligent and adaptive depth sensing. Overall, I believe our method holds great promise for a wide range of applications, such as terrain mapping and autonomous driving.
>
> > **Q3: Limited originality.**
>
> **A3:** Thank you for your attention to this issue. Our work is fundamentally different from [10]. While [10] seeks an optimal coding scheme within the conventional iToF framework, BE-ToF is designed to enable long-distance imaging, which traditional iToF systems cannot support. The double well function loss and first-order difference loss are tailored to our hardware constraints, as non-binary codes are incompatible with our system. Our network employs a lightweight backbone and multi-channel fusion modules, improving both performance and efficiency compared to [10]. We will expand the discussion of BE-ToF in the manuscript to better highlight its contributions.
>
> > **Q4: Limited real ToF dataset experiment.**
>
> **A4:**
>
> 1. Indeed, presenting more real-world experiments and comparisons with different methods is crucial to demonstrating the effectiveness of our BE-ToF system. Therefore, in this rebuttal, we upgrade our hardware setup and conduct experiments across a wider range of scenarios. **The specifications of the upgraded hardware are detailed below:** the system primarily consists of a solid-state laser and an exposure-encodable ICMOS sensor. The laser operates at a wavelength of 532 nm, with a pulse width of 5 ns, a fixed repetition rate of 1 kHz, and a single-pulse energy of 1 mJ. To enable area illumination, we use a laser beam expander. The ICMOS sensor is equipped with a 300–800 mm varifocal lens and a 532 nm band-pass filter (FWHM ±10 nm) to suppress ambient light interference. We conduct experiments across a variety of scenarios, including indoor scenes (\~40 m) such as a hand model and a kettle, as well as outdoor scenes (\~60 m) such as a stone and a staircase. All experiments are conducted under conditions consistent with the simulation setup: we apply the same learned coding function obtained from simulation and use our RSCF-Net for reconstruction. The modulation period $T_m$ is also kept the same as in simulation, set to 20 ns. We conduct a detailed comparison with other methods, and the quantitative results—evaluated using **Mean Absolute Error (MAE) in centimeters**—are presented below. The ground truth is obtained by performing a time-delay scan using our minimum exposure time (3ns), with a step size of 1 ns. The results show that our system consistently achieves centimeter-level depth estimation across various scenarios and outperforms existing methods in terms of accuracy. **We promise to provide reliable visualizations in the camera-ready version.**
>
> |            | Square + PS | Square + RSCF-Net | Learned Code + DeepToF | Learned Code + FisherToF | Ours |
> | :--------: | :---------: | :---------------: | :--------------------: | :----------------------: | :--: |
> | hand model |    10.57    |       3.96        |          4.73          |           2.47           | 1.78 |
> |   kettle   |    9.13     |       3.51        |          4.14          |           1.99           | 1.41 |
> |   stone    |    12.16    |       4.76        |          5.94          |           3.78           | 2.50 |
> | staircase  |    15.79    |       5.92        |          6.70          |           4.26           | 3.83 |
>
> 2. The experiment shown in Fig. 8 of the manuscript follows the same setup as the simulation. The ToF measurements are obtained using the coding functions learned from simulation, and the depth is reconstructed using the pre-trained RSCF-Net.
>
> 3. We sincerely thank the reviewer for recommending these works. Both datasets are of high quality and provide valuable benchmarks for evaluating the performance of our network. However, due to time constraints, we have not yet completed the retraining and evaluation of our network and other comparative methods on these datasets. We will supplement our experimental results at a later stage.
>
> 4. We regret that we cannot modify our system prototype for a direct comparison with FisherToF and DeepToF, as our laser operates at a fixed repetition rate of 1 kHz. Conventional iToF systems rely on high-frequency modulated lasers (tens to hundreds of MHz), which inherently have limited output power and are thus restricted to short-range sensing. In contrast, our BE-ToF system eliminates the need for high modulation frequencies, allowing the use of a low-repetition-rate laser with high-energy pulses, making it well-suited for long-range depth imaging.
>
> > **Q5: Supplementary references in related work.**
>
> **A5:** Thank you for your valuable suggestion. The suggested papers are indeed closely related to our work, and we have now included them in the revised manuscript.

---

> > ### Comment · Reviewer_56Ri · 2025-08-05
> >
> > Thanks for the clarifications and extra experiments
> > Clarifiaction makes sense, while just one minor question regarding that statement in line 189-190 again. [35] doesn't really mention relation regarding "R-channel of the albedo map" and roughness ($\(\rho_s\)$). They are independently obtained from the decomposition, right?

---

> > > ### Author Response · Authors · 2025-08-06
> > >
> > > > **Q1: Clarification on the relationship between "R-channel of the albedo map" and reflectance $\rho_s$.**
> > >
> > > **A1:** Thank you for your concern about this question. **In the Eq. 2 of [35]**, the intrinsic image decomposition model explicitly decomposes the RGB image $I(p)$ into the following components: $I(p) = B(p) \cdot T(p) = S_B(p) \cdot R_B(p) \cdot T(p)$, where $S_B(p)$, $R_B(p)$ and $T(p)$ denote the shading, reflectance and texture components at pixel position $p$, respectively. **In our paper, the "R-channel of the albedo map" corresponds to $R_B(p)$ in the equation above. Since [35] explicitly defines $R_B(p)$ as the scene's reflectance, it is reasonable for us to adopt $R_B(p)$ as the reflectance $\rho_s$ of our paper.** And $R_B(p)$ is obtained independently from the intrinsic image decomposition. We will carefully revise the manuscript to avoid any possible misunderstandings. If you have any other questions, please let us know. We will be happy to clarify them for you.

---

> > > > ### Comment · Reviewer_56Ri · 2025-08-06
> > > >
> > > > R-channel of the albedo sounds really misleading in that case. It rather sounded like, for example albedo[:,:,0] = rho_s, which assigns the red channel of albedo with the reflectance .. So, its just plugging the elements to the right location in the equation.

---

> > > > > ### Author Response · Authors · 2025-08-06
> > > > >
> > > > > We truly appreciate your thoughtful comment and apologize for any confusion caused. **In the camera-ready version, we will carefully clarify our choice of reflectance $\rho_s$ and clearly distinguish it from the red channel.** Your valuable feedback has been instrumental in helping us improve the quality of our manuscript, and we are sincerely grateful.

---

### Official Review · Reviewer_ouhA · 2025-06-28

**Clarity:** 3
**Significance:** 3
**Originality:** 2
**Rating:** 5
**Confidence:** 4

**Summary:**

This paper addresses the problem of long-distance high-fidelity depth imaging with time-of-flight (ToF) systems. Mainly, this work proposes a joint optimization of the coding function with a depth estimation neural network in an end-to-end manner. The work uses a differentiable modeling of the acquisition model, including two regularization functions to achieve implementable coding functions, i) a polynomial function to achieve a square wave, i.e., binary valued functions ii) total variation, so that the coding function does not transition from 1 to 0 so fast. The paper also introduces a restormer-based depth estimation network which includes, beyond the traditional MSE recovery loss function, an information-theory loss function based on the Fisher information matrix to overcome the SNR in the measurements. A comprehensive evaluation of the proposed method, along with SOTA comparison, is provided, showing improvements of the paper in determined distance ranges. This is also validated with an experimental ToF prototype, showcasing the method in real acquisition scenarios.  Supplementary material includes a detailed mathematical derivation of the sensing model and additional results with other datasets.

**Questions:**

1. Why in Figs 5, 6 of the main manuscript and Figs 2 and 3 of the supplementary material not include a quantitative metric to fairly evaluate the quality of the proposed approach?

2. Since for promoting values a degree-two polynomial is used, is it guaranteed that this polynomial achieves exact 0 and 1 values? Why not simply use a binary activation function with a Straight-Through Estimator for the gradient as widely used in binary neural networks? [1]

3. Did you try with fewer measurements? $k<4$?

4. Usually, loss functions are chosen to have range $[0,\infty]$, but the used regularizer has negative values. Why not simply choose a polynomial function $f(x) = (x-1)^2 (x)^2$? Is there any advantage to the proposed approach?

[1]  Le, Huu, et al. "Adaste: An adaptive straight-through estimator to train binary neural networks." Proceedings of the IEEE/CVF Conference on Computer Vision and Pattern Recognition. 2022.

**Ethical Concerns:**

["NO or VERY MINOR ethics concerns only"]

**Final Justification:**

The paper presents a new method for end-to-end learning of coding functions for long-range depth estimation, jointly trained with a depth-estimation network. It proposes a differentiable framework to design binary coding sequences, with carefully designed regularizers that promote physically implementable solutions. Although end-to-end imaging-system design has been widely studied across modalities, this work provides clear design criteria and is well written. The original submission offered a comprehensive evaluation in both simulation and real data, showing improved performance over baselines across three depth ranges. During the rebuttal and discussion period, the authors added important comparisons with state-of-the-art coding-function designs (showing consistent gains), conducted ablation studies on the regularization hyperparameters, and included coding-function optimality metrics based on the Fisher information matrix, improving the interpretability of the learned patterns. I encourage the authors to incorporate these analyses into the manuscript and/or supplementary material. Based on these points, I raise my score to 5 (accept).

**Limitations:**

Although the paper mentions that the method is constrained to a narrow range of distance $\sim 3 $m,  it does not mention how to address those limitations regarding the limited range in future work.

**Paper Formatting Concerns:**

I did not find any paper formatting issues.

**Quality:**

3

**Strengths And Weaknesses:**

Strengths:

1. The proposed method addresses the physical constraints for the coding functions, which are required for implementation. Three different regularizations are proposed to obtain wide binary coding functions and maximize the depth quality.

2. The results are superior to the state of the art in all cases, both quantitatively and qualitatively. The results in terms of MAE are better for different levels of signal-to-noise ratio, which guarantees the robustness to noise and general improvement.

3. The proposed method bridges a gap present in dToF and iToF to achieve accurate and long-distance depth imaging in a limited depth range.

Weaknesses

1. An in-depth analysis of the coefficients of the regularizers ($\gamma_1, \gamma_2, \gamma_3$) used in Eq. (11) is required. Ablation studies and convergence analysis could be added.

2. Although higher quality is achieved for long distances, the range of action is limited by $T_m$. The difference between the minimum and maximum depth is only 3 meters. A discussion of how this could be increased would improve the work.

3. While the method is compared to other state-of-the-art approaches, the evaluation is limited to using their reconstruction networks with the proposed coding functions. A more comprehensive and fair comparison would involve training the coding functions from other methods with the same reconstruction networks used in this paper, to better isolate and assess the contribution of the coding stage itself.

4. There should be an analysis of the “optimality” of the resulting coding functions. Is there a way to interpret them? Additionally, there should be a comparison with other coding functions  [1,2].

[1] Gupta, Mohit, et al. "What are optimal coding functions for time-of-flight imaging?." ACM Transactions on Graphics (TOG) 37.2 (2018): 1-18.
[2] Gutierrez-Barragan, Felipe, et al. "Practical coding function design for time-of-flight imaging." Proceedings of the IEEE/CVF Conference on Computer Vision and Pattern Recognition. 2019.

---

> ### Author Rebuttal · Authors · 2025-07-31
>
> We are sincerely grateful to the reviewer for acknowledging the high imaging quality and strong noise robustness of our work. In this rebuttal, we would like to respond to the following questions raised by the reviewer.
>
> > **Q1: Ablation studies and convergence analysis about coefficients in Eq. 11.**
>
> **A1:** Thank you for your suggestion. In our loss function, MSE serves as the primary term to optimize the network’s depth reconstruction performance, while three additional regularization terms act as auxiliary constraints to guide the learning of an optimal binarized coding function. To validate our coefficient choices, we perform comprehensive ablation studies and present the following conclusions.
>
> **For coefficient $\gamma_1$**, setting it below $5 \times 10^{-6}$ prevents the learned coding function from capturing sufficient Fisher information. Conversely, if it exceeds $5 \times 10^{-3}$, it impairs the dominance of the MSE loss, resulting in slower convergence and reduced reconstruction quality. To balance these effects, we set $\gamma_1$ to $5 \times 10^{-4}$ during the first 40 epochs. After observing a notable decline in the MSE loss beyond 40 epochs, we further reduce it to $5 \times 10^{-5}$ to ensure optimal reconstruction performance.
>
> **For coefficient $\gamma_2$**, setting it below $5 \times 10^{-2}$ prevents the coding function from achieving effective binarization; however, when it exceeds 30, it slows down the network’s convergence. Therefore, we set $\gamma_2$ to $5 \times 10^{-2}$ during the first 40 epochs to allow the network to learn an initially optimal coding function. After epoch 40, we increase it to 1 to enforce strict binarization of the coding function.
>
> **For coefficient $\gamma_3$**, setting it below $5 \times 10^{-2}$ results in a coding function with narrow peaks, which is unsuitable for hardware implementation. On the other hand, values above 10 impose excessive constraints, degrading the reconstruction performance. Therefore, we set $\gamma_3$ to 5 in our network to strike a balance between hardware feasibility and reconstruction quality.
>
> > **Q2: Discussion on how to improve the depth range of BE-ToF system.**
>
> **A2:** We sincerely appreciate your interest in our work. While we acknowledge the current limitations of BE-ToF, we are actively investigating several promising directions to address them in future research. One practical approach leverages BE-ToF’s flexible control over the depth range via time-delay adjustment to perform temporal scanning, thereby synthesizing a comprehensive, wide-range depth map. Considering the high latency potentially introduced by temporal scanning, we propose leveraging compressive sensing to reduce the amount of data acquisition. In addition, a more ideal approach is to adopt a coarse-to-fine strategy: initially capturing a wide-range but low-resolution depth map, followed by selective high-precision depth acquisition in regions of interest (ROIs) using BE-ToF. This approach can be further enhanced with AI-driven techniques to enable intelligent and adaptive depth sensing. Overall, I believe our method holds great promise for a wide range of applications, such as terrain mapping and autonomous driving.
>
> > **Q3: Comparison with other coding functions with the same reconstruction network.**
>
> **A3:** Thank you for your suggestion. This experiment indeed plays a key role in demonstrating the strength of our learnable binarized coding function. To ensure a fair comparison, we employ the same imaging setup (with **K = 4** measurements) and reconstruction network, varying only the coding function. The quantitative results, evaluated using Mean Absolute Error (MAE) in millimeters, are presented below. It can be observed that, under the same reconstruction network, our coding function delivers the best performance. Notably, although we compare multiple coding functions in simulation, only the square wave and our learned binarized coding functions are implementable on actual hardware. The Sinusoid wave cannot be used because it is non-binary, while Hamiltonian codes and M-sequences exhibit narrow peaks that cannot be realized due to hardware constraints on minimum exposure time.
>
> |             |  0-3 m   |  30-33 m  |  60-63 m  |  90-93 m  |
> | :---------: | :------: | :-------: | :-------: | :-------: |
> |  Sinusoid   |  24.67   |   31.12   |   39.80   |   45.39   |
> |   Square    |  16.40   |   22.66   |   26.05   |   33.35   |
> | Hamiltonian |  11.28   |   14.53   |   21.74   |   27.11   |
> | M-sequence  |  15.19   |   21.24   |   28.33   |   35.81   |
> |  **Ours**   | **8.52** | **12.86** | **18.20** | **23.51** |
>
> > **Q4: Analysis of the optimality of the learned coding functions.**
>
> **A4:** Thank you for your valuable suggestion. This is quite important to demonstrate the optimality of our learned coding functions. As discussed in Reference [10] of the manuscript, Fisher information can be used as a metric to assess the optimality of different coding schemes — a higher Fisher information value indicates a more optimal coding scheme. Therefore, we list the Fisher information of different coding functions in the table below for a straightforward comparison. It can be observed that our learnable binarized coding function achieves the highest Fisher information, which demonstrates its optimality.
>
> |  Coding Function   |      Sinusoid      |       Square       |     M-sequence     |    Hamiltonian     |        Ours        |
> | :----------------: | :----------------: | :----------------: | :----------------: | :----------------: | :----------------: |
> | Fisher Information | $1.27 \times 10^6$ | $2.29 \times 10^6$ | $2.18 \times 10^6$ | $2.92 \times 10^6$ | $3.93 \times 10^6$ |
>
> > **Q5: Quantitative metrics for Fig. 5 and 6 in the main manuscript, and Fig. 2 and 3 in the supplementary material.**
>
> **A5:** We sincerely apologize for the misunderstanding caused. In fact, we have conducted detailed quantitative analyses for each experiment. **In the main manuscript**, Fig. 5 primarily compares our BE-ToF method with several traditional iToF approaches, with the corresponding quantitative metrics summarized in Table 1(a). Fig. 6 presents two sets of experiments: the first compares traditional square demodulation functions with our RSCF-Net, with quantitative results shown in Table 1(b); the second compares the performance of different networks using our learned coding functions, with corresponding metrics provided in Table 1(c). **In the supplementary material**, Fig. 1 illustrates the generalization results on the 4D Light Field Dataset compared to traditional iToF methods, with quantitative metrics summarized in Table 1(a). Fig. 2 shows the generalization results on the SUN RGB-D Dataset, with corresponding quantitative metrics provided in Table 1(b). We will revise the manuscript to clearly present the results of each experiment, enhancing readability and avoiding potential misunderstandings.
>
> > **Q6: The optimization precision of the binarized coding functions and the reason for not using a binary activation function with a Straight-Through Estimator.**
>
> **A6:** We appreciate your attention to this issue. Since we employ a differentiable double well function to encourage the coding functions to converge toward 0 or 1, it cannot numerically attain exact binary values. Nevertheless, our examination shows that the learned coding functions are already very close to 0 and 1 (e.g., around 0.0001 or 0.999), and we consider such a small deviation negligible and acceptable for our network. We choose the double well function rather than the Straight-Through Estimator because we aim to implement a **fully differentiable** approach for learning a binarization function, without relying on any gradient approximation. We believe this represents a meaningful and unexplored direction, with our experiments validating the simplicity and effectiveness of the double well function.
>
> > **Q7: Experiments with fewer measurements like $K < 4$.**
>
> **A7:** As noted in lines 135–136 of the manuscript, at least K ≥ 3 measurements are required to recover depth. In our work, we choose K = 4 to ensure high reconstruction quality and robustness to noise. Below, we present quantitative results for cases with K < 4, evaluated using Mean Absolute Error (MAE) in millimeters. When K < 3, the reconstruction quality significantly degrades, which is reasonable given the limited information available for depth recovery. With K = 3, depth can be reasonably reconstructed, though still slightly inferior to K = 4, where the additional measurement improves robustness to noise and other perturbations.
>
> |       | 0-3 m  | 30-33 m | 60-63 m | 90-93 m |
> | :---: | :----: | :-----: | :-----: | :-----: |
> | K = 1 | 133.22 | 166.90  | 156.18  | 198.19  |
> | K = 2 | 37.84  |  42.94  |  51.76  |  49.18  |
> | K = 3 | 14.58  |  17.59  |  21.64  |  29.88  |
> | K = 4 |  8.52  |  12.86  |  18.20  |  23.51  |
>
> > **Q8: Clarification on the choose of the double well function loss.**
>
> **A8:** Thank you for pointing this out. Traditional loss functions like MSE are non-negative because they quantify the error between predictions and ground truth. In contrast, our double well loss serves as a regularization term applied directly to the coding function and is not based on ground truth, so it may take negative values. Moreover, many other loss functions [1,2] also contain negative values, and it has been demonstrated that this does not hinder the network’s optimization process. As for why we don’t use $x^2(x-1)^2$ instead: Although both functions have a similar shape, our double well function exhibits more pronounced gradient variations during [0, 1], which helps the network converge more quickly.
>
>
>
> [1] Adler J, Lunz S. Banach wasserstein gan[J]. Advances in neural information processing systems,2018
>
> [2] Wang Q, Ma Y, Zhao K, et al. A comprehensive survey of loss functions in machine learning[J]. Annals of Data Science,2022

---

> > ### Comment · Reviewer_ouhA · 2025-08-03
> > **Response to rebuttal**
> >
> > Thanks to the authors for the rebuttal. Most of my comments were addressed, however,  I still have some concerns:
> >
> > 1. In **Q1**, I still consider that evaluating the performance of the method under different hyperparameter settings of $\gamma_1,\gamma_2$ and $\gamma_3$ would significantly benefit the understanding of the paper.
> >
> > 2. In the **Q3**, is the reconstruction network the same that was trained jointly with the encoding functions, or was it trained from scratch with the corresponding encoding functions? This is important for a fair comparison of the optimality of the coding function.
> >
> > 3. Regarding **Q6**, what if, in testing, a round function is applied to the learned coding functions to achieve true binary values? Does this significantly reduce performance?

---

> > > ### Author Response · Authors · 2025-08-04
> > >
> > > We sincerely thank the reviewer for the timely and insightful feedback. We provide detailed responses to the reviewer's concerns below.
> > >
> > > > **Q1: Performance of the method under different hyperparameter settings.**
> > >
> > > **A1:** Thank you for your valuable suggestion. We conduct detailed experiments on the hyperparameter settings and summarize the results below. We hope these results provide clarity and aid in understanding our method.
> > >
> > > **For coefficient $\gamma_1$ before epoch 40**, when $\gamma_1$ is below 5e-6, its influence is too weak for the network to learn effective coding functions, leading to performance drop. When it exceeds 5e-4, it disrupts the double-well and first-order difference losses, resulting in coding functions unsuitable for hardware. Thus, we set $\gamma_1$ to 5e-4 during the first 40 epochs.
> > >
> > > | $\gamma_1$ before 40 epoch | 5e-7  | 5e-6  | 5e-5 | 5e-4 |           5e-3           |           5e-2           |
> > > | :------------------------: | :---: | :---: | :--: | :--: | :----------------------: | :----------------------: |
> > > |          MAE (mm)          | 17.23 | 12.47 | 9.76 | 8.52 | Hardware Unimplementable | Hardware Unimplementable |
> > >
> > > **For coefficient $\gamma_1$ in epochs after 40**, we find that the value of $\gamma_1$ has little impact on the final performance. However, setting it too high can slow down the convergence of the network. Therefore, we set $\gamma_1$ to 5e-5 after 40 epochs to balance performance and convergence speed.
> > >
> > > | $\gamma_1$ after 40 epoch | 5e-7 | 5e-6 | 5e-5 | 5e-4  | 5e-3  | 5e-2  |
> > > | :-----------------------: | :--: | :--: | :--: | :---: | :---: | :---: |
> > > |         MAE (mm)          | 9.94 | 8.73 | 8.52 | 14.58 | 11.62 | 10.17 |
> > > |    Convergence Epochs     | 107  | 103  | 112  |  123  |  133  |  137  |
> > >
> > > **For coefficient $\gamma_2$ in epochs before 40**, we set $\gamma_2$ to a small value so that the first-order difference loss dominates and helps suppress narrow peaks. As shown, when $\gamma_2$ exceeds 1, the learned coding functions exhibit narrow peaks and become unsuitable for hardware implementation. Thus, we set $\gamma_2$ to 5e-2 during the first 40 epochs.
> > >
> > > | $\gamma_2$ before 40 epoch | 5e-4  | 5e-3  | 5e-2 | 5e-1  |   1   |            5             |
> > > | :------------------------: | :---: | :---: | :--: | :---: | :---: | :----------------------: |
> > > |          MAE (mm)          | 11.98 | 12.35 | 8.52 | 12.67 | 10.27 | Hardware Unimplementable |
> > >
> > > **For coefficient $\gamma_2$ in epochs after 40**, we increase the value of $\gamma_2$ to encourage the coding functions to converge more rapidly to a binary state. As observed, setting the coefficient below 5e-2 prevents the coding functions from reaching a binary state, while values above 30 cause noticeable performance degradation. Therefore, we set $\gamma_2$ to 1 after 40 epochs.
> > >
> > > | $\gamma_2$ after 40 epoch |           5e-3           | 5e-2  | 5e-1 |  1   |  10  |  20   |  30   |  40   |
> > > | :-----------------------: | :----------------------: | :---: | :--: | :--: | :--: | :---: | :---: | :---: |
> > > |         MAE (mm)          | Hardware Unimplementable | 10.26 | 9.02 | 8.52 | 9.29 | 11.41 | 10.68 | 17.92 |
> > >
> > > **For coefficient $\gamma_3$,** it can be observed that when the coefficient is too small, narrow peaks appear, making hardware unimplementable. Conversely, a large coefficient results in degraded depth reconstruction quality. A balanced performance is achieved with values between 0.05 and 10; we set it to 5 in our experiments.
> > >
> > > | $\gamma_3$ |           5e-4           |           5e-3           | 5e-2 |  1   |  5   |  10  |  20   |  30   |
> > > | :--------: | :----------------------: | :----------------------: | :--: | :--: | :--: | :--: | :---: | :---: |
> > > |  MAE (mm)  | Hardware Unimplementable | Hardware Unimplementable | 9.02 | 8.98 | 8.52 | 8.53 | 16.88 | 23.76 |
> > >
> > > > **Q3: Clarification on the reconstruction network with different coding functions.**
> > >
> > > **A3:** We sincerely apologize for the confusion caused. In our ablation study on the coding functions, **we train our reconstruction network from scratch** with the corresponding coding functions to ensure a fair comparison.
> > >
> > > > **Q6: Performance with true binary values using a round function.**
> > >
> > > **A6:** Thank you for your thoughtful suggestion. As suggested, we apply a round function during testing to convert coding functions into strict binary codes and compare results without it. The quantitative results below show that strict binarization **does not cause performance degradation**; on the contrary, it slightly improves performance.
> > >
> > > |                        | 0-3m | 30-33m | 60-63m | 90-93m |
> > > | :--------------------: | :--: | :----: | :----: | :----: |
> > > | Without Round Function | 8.52 | 12.86  | 18.20  | 23.51  |
> > > |  With Round Function   | 8.51 | 12.78  | 17.97  | 21.97  |

---

> ### Comment · Reviewer_ouhA · 2025-08-05
> **Response to authors**
>
> Thank you to the authors for the detailed response. All of my concerns have been addressed, and the new analyses and results further strengthen the contributions of the paper.

---

> > ### Author Response · Authors · 2025-08-06
> >
> > We sincerely thank you for your valuable and insightful comments, which greatly help improve the quality of our manuscript and inspire the next stage of our work.

---

### Official Review · Reviewer_XQfn · 2025-07-02

**Clarity:** 2
**Significance:** 3
**Originality:** 3
**Rating:** 5
**Confidence:** 3

**Summary:**

The paper introduces Burst-Encodable Time-of-Flight (BE-ToF) imaging, a novel approach for long-distance depth sensing. By transmitting light in burst patterns and capturing the phase shift over an entire burst period, BE-ToF overcomes limitations of traditional iToF systems such as phase ambiguity. The method employs a learnable, end-to-end framework that optimizes optical coding and the neural network for depth reconstruction. The effectiveness of BE-ToF is demonstrated through both simulated experiments and a prototype setup showing superior results in long-range scenarios.

**Questions:**

Could the authors explicitly explain the methodology behind the synthetic long-range indoor dataset? Are conditions such as lighting, materials, and atmospheric factors reflective of outdoor long-range environments? Addressing this clearly would strengthen the confidence in reported results.

Could the authors provide quantitative evaluation results of the hardware prototype using common ground-truth methods (e.g. LiDAR)? Demonstrating quantitative comparisons with previous methods on real-world data would significantly enhance credibility.

Have the authors considered dynamically shifting the time delay to extend the practical depth range for applications like autonomous driving?

Could the authors include experiments demonstrating performance in dynamic or moving scenarios?

Could the authors clarify whether system parameters such as laser power, measurement count (K), and network complexity match those of baseline methods? Including a runtime analysis would help readers assess the practical implications and feasibility of the proposed approach.

**Ethical Concerns:**

["NO or VERY MINOR ethics concerns only"]

**Final Justification:**

The manuscript provides clear and detailed answers to all my concerns, with new experiments, clarifications, and fairer comparisons that strengthen the work. Therefore, I increased my rating to accept.

In the final version, the authors should incorporate the additional results from the rebuttal, particularly the new real-world quantitative and qualitative results across multiple scenes, to highlight the improvements over existing methods.

**Limitations:**

yes

**Paper Formatting Concerns:**

no formatting concerns

**Quality:**

2

**Strengths And Weaknesses:**

Strengths:

The paper proposes a technically novel method for long-range depth imaging by introducing the burst-encodable time-of-flight (BE-ToF) paradigm, enabling phase unwrapping with just a single modulation frequency. This significantly simplifies system complexity. The approach integrates a hardware-aware, end-to-end learning framework optimizing binary code design and depth reconstruction jointly. The authors demonstrate clear performance improvements over baselines on synthetic data. Additionally, comprehensive ablation studies and validation using a real-world prototype indicate promising practical applicability for high-precision long-range depth sensing.


Weaknesses:

The evaluation methodology for indoor long-range scenarios lacks clarity. It appears that the dataset involves artificially extending indoor scenes to represent longer distances, but this is not explicitly stated or adequately explained. It is uncertain whether simulated indoor long-range scenes sufficiently capture conditions typically found in long-range (usually outdoor) scenarios, such as differing ambient illumination, diverse materials, and atmospheric effects. Furthermore, important practical factors like real sensor noise or beam divergence seem unaccounted for. Thus, the strong quantitative results reported might not accurately reflect real-world performance, necessitating additional validation with genuine long-range data.

While the hardware prototype provides a convincing qualitative depth map demonstration for distances between 22–25 meters, it lacks quantitative evaluation against ground-truth data (e.g., LiDAR). Quantitative assessment on real-world scenarios and comparative analysis with existing methods would significantly strengthen the reliability and practical relevance of the results.

Minor:

Optimizing coding and reconstruction through learning follows previous work, limiting novelty primarily to the burst-encodable concept itself.

The stated motivation targets autonomous driving and robotics; however, the depth window presented is narrow. Practical automotive applications would require extending this range dynamically, which is not demonstrated. Additionally, experiments involving dynamic or moving scenes are absent but crucial to understanding potential method limitations.

While hardware specifications are provided, comparisons regarding laser power, measurement number (K), and network complexity against baselines remain unclear. Clarifying these details and including runtime analysis would improve understanding of the method's practical feasibility.

---

> ### Author Rebuttal · Authors · 2025-07-31
>
> We are grateful to the reviewer for recognizing the novelty and promising applicability of our work. In this rebuttal, we would like to offer our responses to the following questions.
>
> > **Q1: The methodology behind the synthetic long-range indoor dataset and the differences compared to long-range outdoor scenarios.**
>
> **A1:** Thank you for point this out. We use the NYU-V2 RGB-D dataset to train and test our network. First, we scale the depth data to a fixed range of 3 meters (corresponding to $T_m$=20ns), which is consistently applied across all simulation experiments to ensure fairness. As analyzed in Sec. 3.1 of the manuscript, the ToF imaging process is mainly influenced by ambient light, scene reflectance, and signal attenuation. To model these factors, we apply intrinsic image decomposition [35] to the RGB images to extract ambient illumination and reflectance components. We also calibrate a signal attenuation curve under long-range conditions within the simulation environment. Since ambient light and scene reflectance are invariant with respect to depth, variations in ToF imaging across distances are primarily attributed to signal attenuation. Therefore, given a constant emitted signal, we apply distance-dependent attenuation coefficients to simulate the reflected signal, which is subsequently used to generate the corresponding ToF measurements. Moreover, sensor noise is explicitly modeled in our simulation as described in Eq. 5, by incorporating both dark current noise and readout noise to produce more realistic ToF measurements. **Our key distinction from outdoor long-range scenarios lies in several aspects:** First, our current simulation does not yet incorporate atmospheric effects, which can significantly influence ToF imaging in outdoor environments. Additionally, beam divergence remains an important factor affecting image quality. To address this, we employ a laser beam expander to generate sheet laser illumination; however, this still deviates somewhat from ideal uniform lighting. In future work, we plan to integrate atmospheric effects into our simulation framework to better align simulated results with real-world experiments. Meanwhile, techniques such as spatial filtering are used to enhance the uniformity of the sheet laser illumination.
>
>
>
> > **Q2: Additional real-world experimental results and quantitative evaluation against ground-truth data.**
>
> **A2:** Indeed, presenting more real-world experiments and comparisons with different methods is crucial to demonstrating the effectiveness of our BE-ToF system. Therefore, in this rebuttal, we upgrade our hardware setup and conduct experiments across a wider range of scenarios. **The specifications of the upgraded hardware are detailed below:** the system primarily consists of a solid-state laser and an exposure-encodable ICMOS sensor. The laser operates at a wavelength of 532 nm, with a pulse width of 5 ns, a fixed repetition rate of 1 kHz, and a single-pulse energy of 1 mJ. To enable area illumination, we use a laser beam expander. The ICMOS sensor is equipped with a 300–800 mm varifocal lens and a 532 nm band-pass filter (FWHM ±10 nm) to suppress ambient light interference.
>
> We conduct experiments across a variety of scenarios, including indoor scenes (\~40 m) such as a hand model and a kettle, as well as outdoor scenes (\~60 m) such as a stone and a staircase. All experiments are conducted under conditions consistent with the simulation setup: we apply the same learned coding function obtained from simulation and use our RSCF-Net for reconstruction. The modulation period $T_m$ is also kept the same as in simulation, set to 20 ns. We conduct a detailed comparison with other methods, and the quantitative results—evaluated using **Mean Absolute Error (MAE) in centimeters**—are presented below. **The ground truth is obtained by performing a time-delay scan using our minimum exposure time (3ns), with a step size of 1 ns.** We do not use LiDAR as the ground truth because LiDAR typically provides sparse point clouds, whereas we obtain dense depth maps. The significant difference between the two makes direct comparison difficult. The results show that our system consistently achieves centimeter-level depth estimation across various scenarios and outperforms existing methods in terms of accuracy. **We promise to provide reliable visualizations in the camera-ready version.**
>
> |            | Square + PS | Square + RSCF-Net | Learned Code + DeepToF | Learned Code + FisherToF | Ours |
> | :--------: | :---------: | :---------------: | :--------------------: | :----------------------: | :--: |
> | hand model |    10.57    |       3.96        |          4.73          |           2.47           | 1.78 |
> |   kettle   |    9.13     |       3.51        |          4.14          |           1.99           | 1.41 |
> |   stone    |    12.16    |       4.76        |          5.94          |           3.78           | 2.50 |
> | staircase  |    15.79    |       5.92        |          6.70          |           4.26           | 3.83 |
>
>
>
>
>
> > **Q3: Potential solutions to extend the practical depth range such as dynamically shifting the time delay.**
>
> **A3:** We sincerely appreciate your interest in our work. While we acknowledge the current limitations of BE-ToF, we are actively investigating several promising directions to address them in future research. One practical approach leverages BE-ToF’s flexible control over the depth range via time-delay adjustment to perform temporal scanning, thereby synthesizing a comprehensive, wide-range depth map. Considering the high latency potentially introduced by temporal scanning, we propose leveraging compressive sensing to reduce the amount of data acquisition. In addition, a more ideal approach is to adopt a coarse-to-fine strategy: initially capturing a wide-range but low-resolution depth map, followed by selective high-precision depth acquisition in regions of interest (ROIs) using BE-ToF. This approach can be further enhanced with AI-driven techniques to enable intelligent and adaptive depth sensing. Overall, I believe our method holds great promise for a wide range of applications, such as terrain mapping and autonomous driving.
>
>
>
>
>
> > **Q4: Experimental performance in dynamic or moving scenes.**
>
> **A4:** We appreciate the reviewer’s visionary suggestions. The ability to handle dynamic scenes is indeed vital for real-world deployment of our BE-ToF system. To evaluate this, we conduct experiments on the 4D Light Field RGB-D dataset. As the dataset contains static scenes, we simulate dynamics by introducing synthetic motion through data augmentation. Quantitative results, measured by Mean Absolute Error (MAE) in millimeters, are shown below. It can be observed that our network exhibits a performance drop in dynamic scenes. This is reasonable cause we have not yet optimized the model for dynamic scenarios. Nevertheless, we recognize the critical importance of dynamic scene depth reconstruction and will prioritize it as a key focus in our future work.
>
> |               | 0-3 m | 30-33 m | 60-63 m | 90-93 m |
> | :-----------: | :---: | :-----: | :-----: | :-----: |
> | Static Scene  | 12.50 |  18.13  |  22.83  |  29.91  |
> | Dynamic Scene | 27.46 |  34.34  |  39.81  |  49.07  |
>
>
>
> > **Q5: Clarification on system parameters such as laser power, measurement count (K), network complexity and runtime.**
>
> **A5:** Thank you for your concern regarding our system. In the manuscript, the laser used is a fiber laser with a wavelength of 905 nm, a repetition rate of 200 kHz, and an average power of 100 mW. In the upgraded hardware described in **A2**, a solid-state laser is employed, featuring a wavelength of 532 nm, a repetition rate of 1 kHz, and an average power of 1 W. Both hardware setups utilize coding functions learned from simulation, with the number of measurements **K = 4**. The comparison of network complexities is summarized in the table below, demonstrating that our RSCF-Net achieves the best reconstruction performance while maintaining relatively low computational complexity and parameter count. Regarding runtime, our system currently runs in offline mode. In the future, we will deploy the algorithm on an FPGA to achieve real-time depth sensing.
>
> |                    | DeepToF | MaskToF | FisherToF | RSCF-Net (Ours) |
> | :----------------: | :-----: | :-----: | :-------: | :-------------: |
> |   **FLOPs (G)**    |  12.65  |  5.62   |   26.72   |      14.69      |
> | **Parameters (M)** |  42.57  |  18.12  |   27.87   |      13.04      |

---

> > ### Comment · Reviewer_XQfn · 2025-08-08
> >
> > The authors’ detailed rebuttal addressed my concerns with clarifications, new experiments, and fairer comparisons, which strengthens the paper and increases my confidence in it.

---

> ### Comment · Area_Chair_x2gb · 2025-08-05
> **Update final rating and justification?**
>
> Dear Reviewer:
>
> Thanks for your review. Could you please take a look at the author response, and update your final rating and justification? Also, please feel free to look at the other reviews while considering your final recommendation...
>
> Best,
>
> Your AC

---

### Official Review · Reviewer_uajT · 2025-07-02

**Clarity:** 4
**Significance:** 4
**Originality:** 2
**Rating:** 5
**Confidence:** 4

**Summary:**

The manuscript proposes a long-distance, high-depth-resolution indirect time-of-flight (iToF) imaging system. The core contribution is the design of learnable modulation and demodulation codes that improve depth resolution, particularly under low and spatially varying signal-to-noise ratios. These codes are burst-type, targeting specific long-distance depth ranges rather than continuous coverage. The authors validate their approach through simulations and demonstrate hardware implementation, presenting a single real-world depth image.

While the proposed method shows promise, I am currently inclined against accepting the manuscript due to (1) lack of simpler and fairer baseline comparisons and (2) lack of more real world results despite the availability of hardware.

**Questions:**

Please answer the questions asked in Strengths And Weaknesses section.

**Ethical Concerns:**

["NO or VERY MINOR ethics concerns only"]

**Final Justification:**

The manuscript provides reasonable answers to all my concerns. Therefore, I increased my rating to accept.

In the final version, the authors should include the new results they provided in rebuttal—especially new qualitative results on multiple scenes showing the improvement over existing codes.

**Limitations:**

Yes

**Quality:**

2

**Strengths And Weaknesses:**

**strengths**
- The manuscript is well written and easy to follow. It builds a hardware prototype for validation.
- Usage of the double well loss function to encourage the decoding function to be binary valued at any time is quite interesting. I haven't seen this loss function before.
- first-order difference loss to avoid narrow peaks (high BW) is also interesting. However, why not sample the D(t) functions at the fastest switching rates possible by hardware and thereby avoid this loss function altogether. Also, loss functions encourage a behaviour but do not guarantee that it will happen. How is that handled?
**weaknesses**
- While the proposed technique enhances long-distance sensing, it does not extend the depth range---unlike dual-modulation, frequency switching, or M-sequence-based methods, which are specifically designed for that purpose. As a result, the current comparisons may be somewhat unfair to these techniques, as they are evaluated on metrics outside their intended strengths.
-  A more fair and complete comparison---currently missing---is an ablation study where a single-frequency waveform (e.g., a sinusoid with a frequency chosen to match the same depth range as used in this manuscript) is employed, but temporally delayed by appending zeros. This raises the question: is the use of neural-optimized codes truly necessary to achieve good performance?
- Comparison with space filling codes (optimal codes for ToF imaging [1]) or M-sequences [2] which can also lead to long distance (and range) is missing.
     - [1] Gupta, M., Velten, A., Nayar, S.K. and Breitbach, E., 2018. What are optimal coding functions for time-of-flight imaging?. ACM Transactions on Graphics (TOG), 37(2), pp.1-18.
     - [2] Kadambi, A., Whyte, R., Bhandari, A., Streeter, L., Barsi, C., Dorrington, A. and Raskar, R., 2013. Coded time of flight cameras: sparse deconvolution to address multipath interference and recover time profiles. ACM Transactions on Graphics (ToG), 32(6), pp.1-10.
- While the manuscript builds the hardware, it includes only a single experimental result, even in the supplementary material. Moreover, the result lacks quantitative or qualitative comparisons with alternative coding strategies. Capturing and evaluating multiple scenes under varying conditions---such as different times of day---and comparing the performance across different codes would significantly strengthen the manuscript and provide a more comprehensive validation of the proposed system.

- Eq. 5 is inaccurate. The light itself is Poisson distributed. Given that the system is operating in low-light conditions, this is quite important.
   - I am at Eq. 9 and noticed that light is considered Poisson accurately here. So, it is just a typo in Eq. 5. Please fix it.

**Composition: **
-  Equations are part of the text and should therefore include appropriate punctuation. Specifically, Eqs. 1–5, 7, 9, and 11 should end with commas, while Eqs. 6, 9, and 10 should end with periods.
- Lines 133–134: Names derived from proper nouns must be capitalized. Use 'Poisson' and 'Gaussian' instead of 'poisson' and 'gaussian'.
- Move Fig 5 to top of the page as well, like the rest of the figures. If not, it comes after the text and appears like it is part of the text.

---

> ### Author Rebuttal · Authors · 2025-07-31
>
> We sincerely appreciate the reviewer’s recognition of our paper as well written and interesting. In this rebuttal, we would like to provide our responses to the following questions.
>
> >**Q1: The necessity of first-order difference loss and how to guarantee the avoidance of narrow peaks.**
>
> **A1:** Thank you for your interest in this question. Since our coding functions are primarily used to modulate the sensor’s exposure, they are constrained by the minimum exposure time achievable by the hardware, which is currently limited to **3 ns**. This restriction makes it infeasible to finely sample the coding functions to avoid first-order difference loss altogether. The avoidance of narrow peaks is guaranteed by adjusting the loss balance coefficient $\gamma_3$ in Eq. 11. To validate the appropriateness of our coefficient selection, we present an ablation study on $\gamma_3$ below. It can be observed that when the coefficient is too small, it fails to avoid the appearance of narrow peaks, while a large coefficient leads to degraded depth reconstruction quality. A relatively balanced performance can be achieved when the coefficient is chosen within the range of 0.05 to 10. In our experiments, we set it to 5.
>
> | $\gamma_3$ |          0.0005          |          0.005           |    0.05     |      1      |      5      |     10      |     20      |     30      |
> | :--------: | :----------------------: | :----------------------: | :---------: | :---------: | :---------: | :---------: | :---------: | :---------: |
> |   Result   | Hardware Unimplementable | Hardware Unimplementable | Appropriate | Appropriate | Appropriate | Appropriate | Low Quality | Low Quality |
>
> > **Q2: Ablation study on coding strategies and the necessity of optimizing the coding function via neural networks.**
>
> **A2:** Thank you for your valuable suggestion. Our comparison with the dual-frequency modulated iToF system in the manuscript is primarily intended to highlight the advantages of our BE-ToF system—achieving high-precision, long-distance depth imaging using only a single modulation frequency. To ensure a fair comparison of coding functions, we employ the same imaging setup (with **K = 4** measurements) and reconstruction network, varying only the coding function. The quantitative results, evaluated using Mean Absolute Error (MAE) in millimeters, are presented below. It can be observed that, under the same reconstruction network, our coding function delivers the best performance. Notably, although we compare multiple coding functions in simulation, **only the square wave and our learned binarized coding function are implementable on actual hardware.** The sinusoidal wave is excluded due to its non-binary nature, while Hamiltonian codes and M-sequences contain narrow peaks that are impractical to implement given hardware constraints on minimum exposure time.
>
> |             |  0-3 m   |  30-33 m  |  60-63 m  |  90-93 m  |
> | :---------: | :------: | :-------: | :-------: | :-------: |
> |  Sinusoid   |  24.67   |   31.12   |   39.80   |   45.39   |
> |   Square    |  16.40   |   22.66   |   26.05   |   33.35   |
> | Hamiltonian |  11.28   |   14.53   |   21.74   |   27.11   |
> | M-sequence  |  15.19   |   21.24   |   28.33   |   35.81   |
> |  **Ours**   | **8.52** | **12.86** | **18.20** | **23.51** |
>
> To further demonstrate the advantages of optimizing coding functions using neural networks, we use **Fisher information** to evaluate the quality of each coding function. As discussed in Reference [10] of the manuscript, Fisher information can be used as a metric to assess the optimality of different coding schemes — a higher Fisher information value indicates a more optimal coding scheme. Therefore, we list the Fisher information of different coding functions in the table below for a straightforward comparison. It can be observed that our learnable binarized coding function achieves the highest Fisher information, which demonstrates its optimality.
>
> |  Coding Function   |      Sinusoid      |       Square       |     M-sequence     |    Hamiltonian     |        Ours        |
> | :----------------: | :----------------: | :----------------: | :----------------: | :----------------: | :----------------: |
> | Fisher Information | $1.27 \times 10^6$ | $2.29 \times 10^6$ | $2.18 \times 10^6$ | $2.92 \times 10^6$ | $3.93 \times 10^6$ |
>
> > **Q3: Comparison with space filling codes and M-sequences.**
>
> **A3:** Thank you for your valuable recommendation and these are indeed two very important works. We provide a detailed comparison of the two coding functions in **A2**, using the same reconstruction network and replacing only the coding function to ensure a fair comparison. The results show that our learnable binarized coding function is not only hardware-implementable but also delivers the best performance.
>
> > **Q4: Additional real-world experimental results.**
>
> **A4:** Indeed, presenting more real-world experiments and comparisons with different methods is crucial to demonstrating the effectiveness of our BE-ToF system. Therefore, in this rebuttal, we upgrade our hardware setup and conduct experiments across a wider range of scenarios. **The specifications of the upgraded hardware are detailed below:** the system primarily consists of a solid-state laser and an exposure-encodable ICMOS sensor. The laser operates at a wavelength of 532 nm, with a pulse width of 5 ns, a fixed repetition rate of 1 kHz, and a single-pulse energy of 1 mJ. To enable area illumination, we use a laser beam expander. The ICMOS sensor is equipped with a 300–800 mm varifocal lens and a 532 nm band-pass filter (FWHM ±10 nm) to suppress ambient light interference.
>
> We conduct experiments across a variety of scenarios, including indoor scenes (\~40 m) such as a hand model and a kettle, as well as outdoor scenes (\~60 m) such as a stone and a staircase. All experiments are conducted under conditions consistent with the simulation setup: we apply the same learned coding function obtained from simulation and use our RSCF-Net for reconstruction. The modulation period $T_m$ is also kept the same as in simulation, set to 20 ns. We conduct a detailed comparison with other methods, and the quantitative results—evaluated using **Mean Absolute Error (MAE) in centimeters**—are presented below. The ground truth is obtained by performing a time-delay scan using our minimum exposure time (3ns), with a step size of 1 ns. The results show that our system consistently achieves centimeter-level depth estimation across various scenarios and outperforms existing methods in terms of accuracy. **We promise to provide reliable visualizations in the camera-ready version.**
>
> |            | Square + PS | Square + RSCF-Net | Learned Code + DeepToF | Learned Code + FisherToF | Ours |
> | :--------: | :---------: | :---------------: | :--------------------: | :----------------------: | :--: |
> | hand model |    10.57    |       3.96        |          4.73          |           2.47           | 1.78 |
> |   kettle   |    9.13     |       3.51        |          4.14          |           1.99           | 1.41 |
> |   stone    |    12.16    |       4.76        |          5.94          |           3.78           | 2.50 |
> | staircase  |    15.79    |       5.92        |          6.70          |           4.26           | 3.83 |
>
> > **Q5: Clarification on Eq. 5.**
>
> **A5:** We appreciate your careful and thorough feedback. In our BE-ToF imaging model, the light signal $I_i(s)$ in Eq. 5 **indeed follows the Poisson distribution**, expressed as $I_i(s) \sim \mathcal{P}(\mathbb{E}[I_i(s)])$. Regarding $n_d$ and $n_r$, they correspond to the dark current noise and readout noise of the camera sensor, which follow Poisson and Gaussian distributions, respectively. Both noise components are incorporated into the imaging model to more accurately simulate real-world measurements. We apologize for any misunderstanding caused. We will carefully revise the description of Eq. 5 to clarify the meaning of each variable.
>
> > **Q6: Writing errors and formatting adjustments.**
>
> **A6:** We sincerely thank you for your expert and insightful suggestions, which have greatly enhanced the clarity and quality of our manuscript. In response, we have carefully revised the equations, figures, and proper nouns as you recommended to further improve the overall readability.

---

> > ### Comment · Reviewer_uajT · 2025-08-05
> > **Regarding simpler comparisons**
> >
> > Can you answer this one?
> >
> > "While the proposed technique enhances long-distance sensing, it does not extend the depth range---unlike dual-modulation, frequency switching, or M-sequence-based methods, which are specifically designed for that purpose. As a result, the current comparisons may be somewhat unfair to these techniques, as they are evaluated on metrics outside their intended strengths.  -A more fair and complete comparison---currently missing---is an ablation study where a single-frequency waveform (e.g., a sinusoid with a frequency chosen to match the same depth range as used in this manuscript) is employed, but temporally delayed by appending zeros. This raises the question: is the use of neural-optimized codes truly necessary to achieve good performance?"

---

> > > ### Author Response · Authors · 2025-08-06
> > >
> > > We sincerely apologize for any inconvenience caused. In fact, we discuss this issue in **A2** of the rebuttal. As you suggested, we have adopted a fairer and more comprehensive comparison. We use different coding functions at a single frequency and implement temporal delays by appending zeros. To ensure fairness, both the reconstruction network and measurements (K=4) are kept consistent. The quantitative results, evaluated using Mean Absolute Error (MAE) in millimeters, are presented below. It can be observed that, under the same reconstruction network, **our learnable coding functions achieve state-of-the-art performance.**
> > >
> > > |             |  0-3 m   |  30-33 m  |  60-63 m  |  90-93 m  |
> > > | :---------: | :------: | :-------: | :-------: | :-------: |
> > > |  Sinusoid   |  24.67   |   31.12   |   39.80   |   45.39   |
> > > |   Square    |  16.40   |   22.66   |   26.05   |   33.35   |
> > > | Hamiltonian |  11.28   |   14.53   |   21.74   |   27.11   |
> > > | M-sequence  |  15.19   |   21.24   |   28.33   |   35.81   |
> > > |  **Ours**   | **8.52** | **12.86** | **18.20** | **23.51** |
> > >
> > > To further highlight the advantages of optimizing coding functions via neural networks, we evaluate the quality of each coding function using **Fisher Information**. As discussed in Reference [10] of the manuscript, Fisher Information serves as a metric for assessing the optimality of different coding schemes — a higher Fisher Information value indicates a more optimal coding scheme. Accordingly, we present the Fisher Information of various coding functions in the table below for straightforward comparison. It can be observed that our learnable coding functions yield the highest Fisher Information, further demonstrating their optimality.
> > >
> > > |  Coding Function   |      Sinusoid      |       Square       |     M-sequence     |    Hamiltonian     |        Ours        |
> > > | :----------------: | :----------------: | :----------------: | :----------------: | :----------------: | :----------------: |
> > > | Fisher Information | $1.27 \times 10^6$ | $2.29 \times 10^6$ | $2.18 \times 10^6$ | $2.92 \times 10^6$ | $3.93 \times 10^6$ |
> > >
> > > We sincerely appreciate your thoughtful attention to this issue. If you have any further questions or concerns, please feel free to let us know. We would be very happy to provide any clarification you may need.

---

### Note · Authors · 2025-08-12

Our work presents a novel BE-ToF system capable of delivering long-distance, high-precision, array-based depth imaging. It addresses the challenges of high hardware demands and sparse depth maps typically associated with dToF, as well as the phase wrapping issues encountered in long-range iToF imaging. We adopt an end-to-end approach to jointly optimize the coding functions and the reconstruction network, introducing novel loss functions to ensure the hardware implementability of the learned coding functions. This technology holds significant potential for applications in fields such as autonomous driving and terrain mapping.



During the rebuttal and discussion, the reviewers primarily focus on the following issues: how to address the limited sensing range of BE-ToF, the limited real-world results presented in the paper, and the absence of comparisons with other coding functions. In response, we provide a detailed discussion on strategies to extend the sensing range of BE-ToF, and we supplement additional real-world results as well as comparisons with other coding functions. The reviewers also raise minor questions regarding the choice of the double-well function loss and first-order difference loss, the synthesis of the dataset, network design, and hyperparameter ablation. For these points, we offer detailed explanations, clarifying the rationale behind our choice of loss functions, the method used for dataset synthesis, and the guiding principles of our network design. Furthermore, we conduct extensive ablation experiments to validate the soundness of our hyperparameter choices. Ultimately, the reviewers unanimously agree that our responses address their concerns, and we are grateful for their recognition.



Finally, we would like to express our heartfelt gratitude to the reviewers for their insightful and constructive comments, which have helped us identify and address many potential issues in our work and have significantly enhanced the quality of our manuscript. We are also deeply grateful to every PC, SAC, and AC for their dedication and hard work throughout the review process, whose thoughtful efforts are essential to upholding the high standards and overall excellence of the conference.

---

### Decision · Program_Chairs · 2025-09-17

**Decision:**

Accept (spotlight)

**Comment:**

This paper received four positive reviews, with all four reviewers converging to accept recommendations --- 4 Accepts.

There was general appreciation for the significance of the problem considered here, the novelty of the solution, the strength of the results and the overall presentation.

While there were some concerns raised in the original reviews, they were adequately addressed during the author-reviewer discussion phase. As a result, an accept decision was reached.